# The herpesviral antagonist m152 reveals differential activation of STING-dependent IRF and NF-κB signaling and STING's dual role during MCMV infection

Markus Stempel[1] , Baca Chan[1], Vanda Juranić Lisnić[2], Astrid Krmpotić[2], Josephine Hartung[1], Søren R Paludan[3] , Nadia Füllbrunn[1], Niels AW Lemmermann[4] & Melanie M Brinkmann[1,5,*]

## Abstract

Cytomegaloviruses (CMVs) are master manipulators of the host immune response. Here, we reveal that the murine CMV (MCMV) protein m152 specifically targets the type I interferon (IFN) response by binding to stimulator of interferon genes (STING), thereby delaying its trafficking to the Golgi compartment from where STING initiates type I IFN signaling. Infection with an MCMV lacking m152 induced elevated type I IFN responses and this leads to reduced viral transcript levels both *in vitro* and *in vivo*. This effect is ameliorated in the absence of STING. Interestingly, while m152 inhibits STING-mediated IRF signaling, it did not affect STING-mediated NF-κB signaling. Analysis of how m152 targets STING translocation reveals that STING activates NF-κB signaling already from the ER prior to its trafficking to the Golgi. Strikingly, this response is important to promote early MCMV replication. Our results show that MCMV has evolved a mechanism to specifically antagonize the STING-mediated antiviral IFN response, while preserving its pro-viral NF-κB response, providing an advantage in the establishment of an infection.

**Keywords** herpesvirus; innate immunity; IRF3; NF-κB; STING
**Subject Categories** Immunology; Microbiology, Virology & Host Pathogen Interaction
**The EMBO Journal (2019) 38: e100983**

## Introduction

Host defense against infection requires the early recognition of invading pathogens by pattern recognition receptors (PRR). DNA derived from pathogens, such as DNA viruses, is a potent pathogen-associated molecular pattern (PAMP), which can be detected by DNA sensors and thereby trigger the production of type I interferons (IFN) and proinflammatory cytokines. Although several DNA sensors have been described, the cyclic guanosine monophosphate-adenosine monophosphate (cGAMP) synthase (cGAS) is considered the major sensor of cytosolic DNA (Sun *et al*, 2013). DNA binding to cGAS leads to the production of the second messenger 2′3′-cGAMP, which then directly binds to the endoplasmic reticulum (ER)-resident protein stimulator of interferon genes (STING) (Ishikawa & Barber, 2008). STING is composed of four transmembrane domains and a long cytoplasmic C terminus (Ouyang *et al*, 2012). Upon activation, STING undergoes dimerization via its C-terminal domain and then translocates from the ER to the Golgi compartment, where it binds to and is phosphorylated by the TANK-binding kinase 1 (TBK1) leading to phosphorylation and activation of the transcription factor interferon regulatory factor 3 (IRF3) and type I IFN transcription (Liu *et al*, 2015). Moreover, STING can also activate nuclear factor kappa-light-chain-enhancer of activated B cell (NF-κB)-dependent signaling; however, the exact mechanism and subcellular compartment from where this signaling pathway is activated remains poorly understood. Previous findings suggest that STING activates canonical and non-canonical NF-κB activation via the TNF receptor associated factor 6 (TRAF6)-TBK1 axis and TRAF3, respectively (Abe & Barber, 2014).

STING is essential for the innate immune response to a variety of viral pathogens. Herpes simplex virus type 1 (HSV-1) was the first DNA virus reported to induce the cGAS-STING pathway (Li *et al*, 2013). Mice lacking cGAS or STING were shown to be susceptible to HSV-1 infection (Reinert *et al*, 2016). Similar observations were made for several other herpesviruses such as Kaposi's sarcoma-associated herpesvirus (KSHV) (Ma *et al*, 2015), human cytomegalovirus (HCMV) (Paijo *et al*, 2016), and murine cytomegalovirus

1  Viral Immune Modulation Research Group, Helmholtz Centre for Infection Research, Braunschweig, Germany
2  Center for Proteomics, Faculty of Medicine, University of Rijeka, Rijeka, Croatia
3  Department of Biomedicine, Aarhus Research Center for Innate Immunology, University of Aarhus, Aarhus, Denmark
4  Institute for Virology and Research Center for Immunotherapy, University Medical Center of the Johannes Gutenberg-University Mainz, Mainz, Germany
5  Institute of Genetics, Technische Universität Braunschweig, Braunschweig, Germany
   *Corresponding author. Tel: +49 531 6181 3069; E-mail: m.brinkmann@tu-bs.de

(MCMV) (Lio *et al*, 2016; Chan *et al*, 2017). The initial burst of type I IFN production upon MCMV infection was shown to be dependent on STING (Lio *et al*, 2016). It follows then that herpesviruses would have evolved discreet mechanisms to overcome this pathway, which is an important source of the potent antiviral type I IFN response.

Through millions of years of co-evolution, herpesviruses have developed effective strategies to moderate immune control for securing lifelong persistence in their respective hosts. In the case of CMV, viral immune evasion of natural killer (NK) cell- and T cell-mediated responses has been identified and well characterized (Lemmermann *et al*, 2012; Lisnic *et al*, 2015). However, the mechanism by which CMV evades innate immune control following PRR signaling remains poorly understood. MCMV is a well-established model to study the delicate balance between CMV and its host. So far, no MCMV protein has been identified to specifically target the cGAS-STING pathway, and no study has described the *in vivo* influence of a beta-herpesviral cGAS-STING modulator.

Here, we describe m152 as the first MCMV protein to specifically engage the adaptor protein STING within the first few hours of infection. m152, which is an ER-resident type I transmembrane protein, has been previously reported to efficiently thwart both NK- and T cell-dependent immune responses by preventing cell surface expression of the NKG2D ligand retinoic acid early inducible gene-1 (RAE-1) and major histocompatibility complex class I molecules (MHC class I), respectively (Ziegler *et al*, 1997; Krmpotic *et al*, 1999; Lodoen *et al*, 2003; Fink *et al*, 2013). We now show that m152 additionally modulates the cGAS-STING pathway, independently of its effect on NK- and T cell-mediated responses. At a very early time point after MCMV infection, m152 perturbs the translocation of activated STING from the ER to the Golgi compartment and thereby inhibits the type I IFN response to MCMV infection. Interestingly, m152 has no effect on STING-mediated NF-κB activation, which suggests that STING may activate NF-κB signaling prior to trafficking. We observed both *in vitro* and *in vivo* that the inhibitory effect of m152 generates a permissive environment resulting in enhanced viral transcription. However, the absence of STING does not create an advantage for MCMV replication in the first hours of infection, which suggests that STING may have a pro-viral role. We made use of the ability of m152 to selectively delay STING translocation from the ER to the Golgi to show that STING activates NF-κB signaling already from the ER and that this response is indeed beneficial for early MCMV transcription. This study highlights a dual role for STING in the context of MCMV infection, as well as the resourcefulness of MCMV in encoding a single viral protein targeting three major immune responses to foster an optimal environment for establishing a successful infection in the host.

# Results

### The MCMV m152 protein specifically downmodulates STING-dependent type I IFN induction

Recently, it was shown that the initial type I IFN response upon MCMV infection depends on the key adaptor protein STING (Lio *et al*, 2016), which mediates signaling downstream of cytosolic DNA sensing. Since MCMV has evolved a plethora of evasion strategies to modulate innate and adaptive immune responses, we hypothesized that MCMV would have evolved a mechanism to counteract the STING-mediated innate immune response. To address this, we developed an unbiased luciferase-based reporter assay in 293T cells to screen for modulators of IFNβ transcription encoded by MCMV. As 293T cells do not express endogenous cGAS or STING, we reconstituted the pathway by transiently expressing a murine Cherry-STING fusion protein and induced signaling by co-expression of cGAS-GFP. To monitor IFNβ induction, a reporter plasmid composed of the murine IFNβ promoter upstream of the firefly luciferase gene (IFNβ-Luc) was co-transfected. In total, 173 MCMV open reading frames (ORFs) (Munks *et al*, 2006) were tested for their ability to inhibit the cGAS-STING pathway (Appendix Fig S1). Among the MCMV proteins tested, the MCMV type I transmembrane protein m152 significantly inhibited IFNβ promoter activity downstream of cGAS-STING signaling to a similar extent as the known IRF3 antagonist KSHV ORF36 (Hwang *et al*, 2009) compared to either empty vector (ev) or the cellular type I transmembrane protein CD4 (Fig 1A). To investigate whether m152 targets multiple PRR-mediated signaling pathways, we co-expressed RIG-I N, a constitutively active form of the cytosolic RNA sensor RIG-I. As expected, influenza NS1, an antagonist of RIG-I signaling, markedly inhibited IFNβ promoter induction, whereas m152, as for the control CD4, had no effect on RIG-I signaling (Fig 1B). Upon overexpression of TBK1 or expression of a constitutively active form of IRF3 (IRF3-5D) (Lin *et al*, 1998), m152 expression had no effect on type I IFN induction (Fig 1C and D). Additionally, to exclude an effect of m152 on interferon-α/β receptor (IFNAR) signaling, 293T cells were co-transfected with an ISG56 promoter firefly luciferase reporter construct and IFNAR signaling was activated by the addition of recombinant IFNβ. MCMV M27, a known modulator downstream of IFNAR-dependent signaling (Zimmermann *et al*, 2005), inhibited ISG56 promoter induction, whereas m152 did not have an effect (Fig 1E). Thus far, these results suggest that the MCMV m152 protein specifically targets STING-dependent signaling prior to the activation of the kinase TBK1 and the transcription factor IRF3.

To validate our results in a more physiological setting, we examined the effect of m152 on IFNβ transcription in murine embryonic fibroblasts (MEF). For this, we utilized immortalized *goldenticket* MEF (iMEF$^{gt/gt}$), which do not express endogenous STING due to an $I_{199}N$ missense mutation in STING (Sauer *et al*, 2011; Appendix Fig S2A). We reconstituted STING expression in these cells by transducing them with a murine Cherry-STING fusion construct resulting in iMEF$^{gt/gt}$ Cherry-STING. These cells were additionally transduced to stably express V5-tagged m152, and the expression was verified by immunoblotting (Appendix Fig S2A). Upon stimulation with interferon-stimulatory DNA (ISD), we observed an inhibition of IFNβ transcription in m152 expressing cells (Fig 1F). In contrast, upon stimulation with the RLR ligand poly(I:C), the presence of m152 did not affect IFNβ transcription (Fig 1G).

Since the cGAS-STING signaling pathway is crucial to mount a potent type I IFN response upon MCMV infection in macrophages (Chan *et al*, 2017), we addressed whether m152 expression reduces IFNβ secretion in this cell type. We generated immortalized bone marrow-derived macrophages (iBMDM) stably expressing V5-tagged m152 (Appendix Fig S2B). As expected, upon stimulation with 2′3′-cGAMP or ISD, lower levels of secreted IFNβ were detected in the presence of m152 (Fig 1H and I). In contrast, m152 had no effect

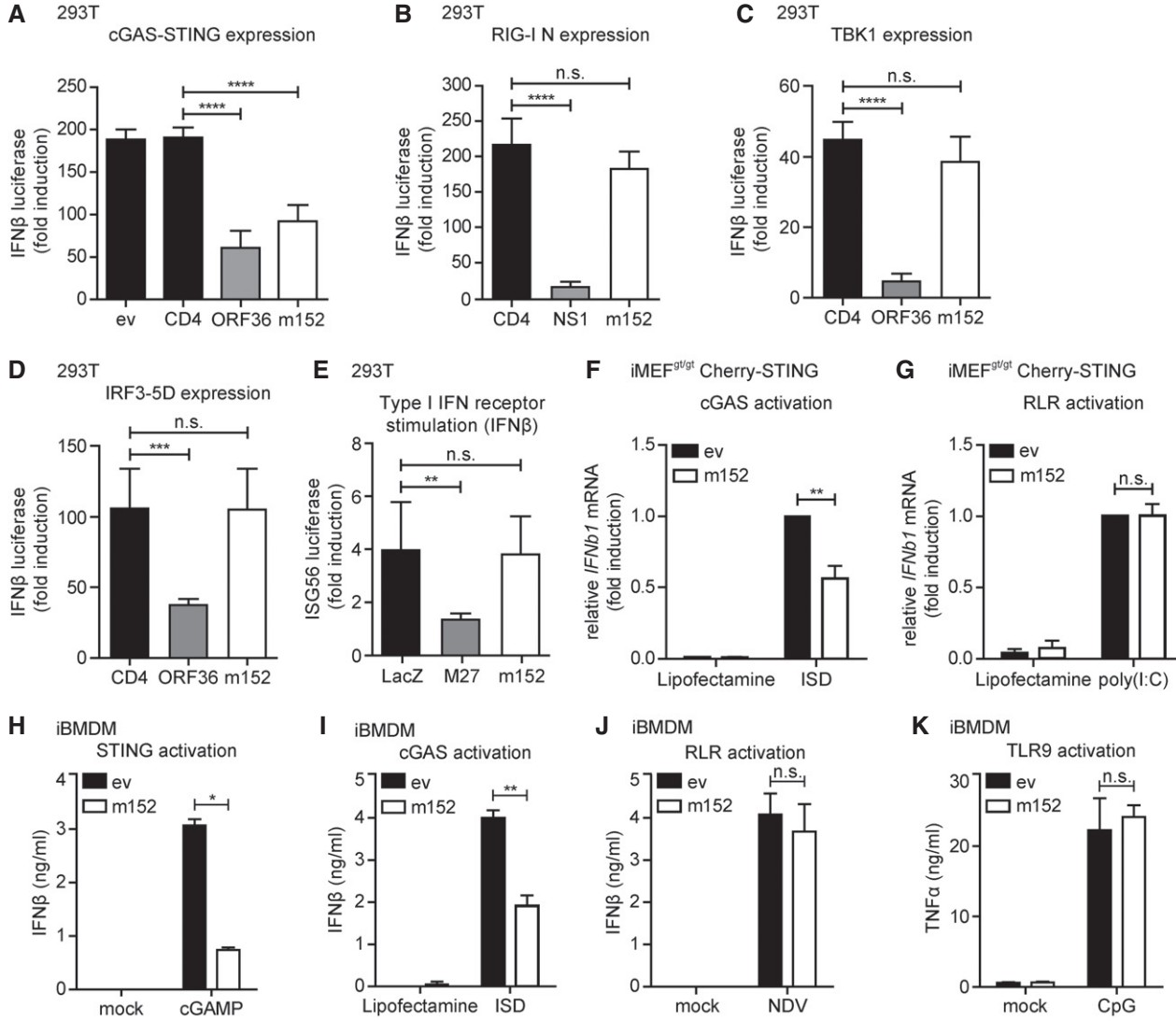

**Figure 1.   The MCMV m152 protein specifically targets STING-dependent signaling.**

A   293T cells were co-transfected with expression plasmids for Cherry-STING, the murine IFNβ-luciferase reporter (IFNβ-Luc), a Renilla luciferase normalization control (pRL-TK), and the indicated expression plasmids or empty vector (ev). Cells were additionally co-transfected with expression plasmids for cGAS-GFP (stimulated) or IRES-GFP (unstimulated). 20 hours post-transfection, cells were lysed and a dual-luciferase assay was performed.

B   An expression plasmid for RIG-I N (stimulated) or ev (unstimulated) was co-transfected with IFNβ-Luc, pRL-TK and the indicated expression plasmids in 293T cells and analyzed as in (A).

C   An expression plasmid for TBK1 (stimulated) or ev (unstimulated) was co-transfected together with IFNβ-Luc, pRL-TK and the indicated expression plasmids and analyzed as in (A).

D   293T cells were co-transfected with a plasmid expressing constitutively active IRF3 (IRF3-5D; stimulated) or IRES-GFP (unstimulated) together with IFNβ-Luc, pRL-TK and the indicated expression plasmids and analyzed as in (A).

E   The ISG56-luciferase reporter, pRL-TK, and the indicated expression plasmids were co-transfected in 293T cells. 24 hours post-transfection, cells were stimulated with 0.1 ng/µl human IFNβ or mock stimulated and analyzed 16 h later as described in (A).

F, G   iMEF^gt/gt stably expressing Cherry-STING and either ev or V5-tagged m152 were stimulated with 5 µg/ml ISD (F), 10 µg/ml poly(I:C) (G), or mock stimulated with Lipofectamine. 4 hours post-stimulation, RNA was extracted to determine IFNβ mRNA transcripts by qRT–PCR.

H–K   iBMDM stably expressing ev or m152-V5 were stimulated in duplicates with 10 µg/ml cGAMP (H), 5 µg/ml ISD (I), Newcastle disease virus (NDV) infection (J), or 1 µM CpG DNA (K). 6 (H) or 16 (I-K) hours later, secreted IFNβ (H-J) or TNFα (K) levels were determined by ELISA.

Data information: (A-G) Data are combined from three independent experiments. (H-K) Experiments were performed three (H, I, K) or two (J) times independently and one representative experiment is shown. Student's *t*-test (unpaired, two-tailed), n.s. not significant, *$P < 0.05$, **$P < 0.01$, ***$P < 0.001$, ****$P < 0.0001$. Data are shown as mean ± SD.

on Newcastle disease virus (NDV)-induced RIG-I signaling (Yoneyama *et al*, 2004) (Fig 1J), nor did it affect CpG-induced TLR9 signaling (Fig 1K). Collectively, these data show that m152 selectively targets cGAS-STING signaling upstream of the TBK1-IRF3 axis, but does not affect RIG-I-, TLR9-, or IFNAR-mediated signaling.

## m152 co-localizes and interacts with STING in resting and stimulated cells

In unstimulated cells, STING is localized in the ER (Ouyang *et al*, 2012). Upon activation, STING translocates from the ER to the Golgi compartment, which is a crucial prerequisite for downstream signaling leading to the induction of the type I IFN response mediated by TBK1 and IRF3 (Liu *et al*, 2015).

MCMV m152 is likewise described as an ER-resident protein (Ziegler *et al*, 1997) and as we have observed that m152 targets STING-mediated signaling, we sought to examine whether m152 translocates together with STING. For this, we first transfected HeLa cells with expression constructs for murine Cherry-STING and V5-tagged m152.

As expected, STING and m152 co-localized in the ER in unstimulated cells (Fig 2A, upper panel). Upon overexpression of cGAS-GFP to activate STING-dependent signaling, we observed that STING translocated to the perinuclear region as described previously (Ishikawa *et al*, 2009) (Fig 2A, lower panel). Notably, m152 co-localized with STING in the perinuclear region. This shows that, like STING, m152 translocates upon stimulation of the cGAS-STING pathway.

Subsequently, we wanted to address whether the translocation of m152 upon cGAS activation is STING-dependent. When transfecting iMEF[gt/gt] with an expression plasmid for V5-tagged m152 alone, m152 was detected in the ER (Fig 2B, first panel). When we reconstituted STING expression, m152 and STING co-localized in the ER and upon overexpression of cGAS, both proteins translocated to the

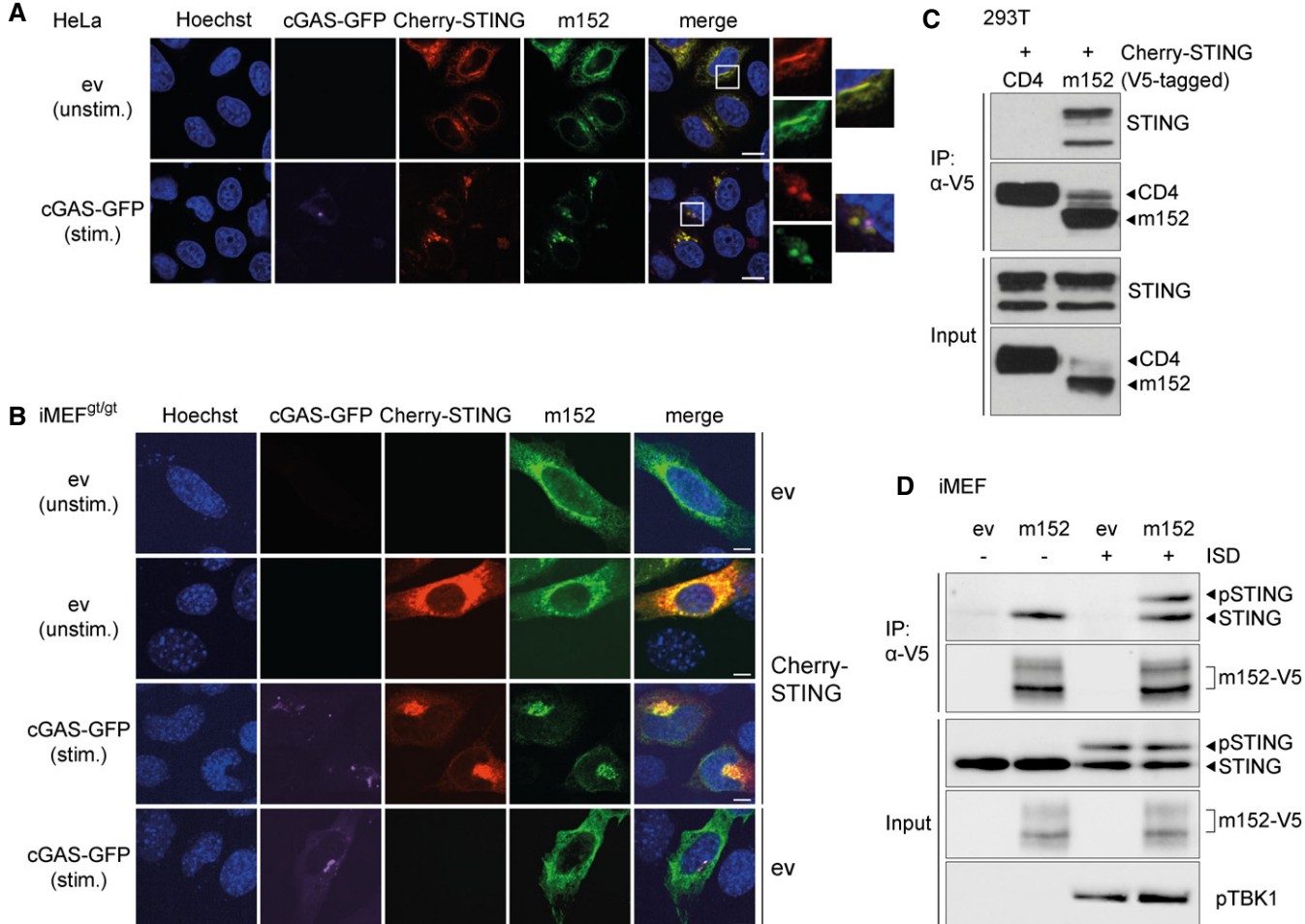

**Figure 2.  STING and the MCMV m152 protein co-localize and interact under unstimulated and stimulated conditions.**

A, B  (A) HeLa cells were co-transfected with expression plasmids for Cherry-STING, V5-tagged m152, and either ev (unstimulated) or cGAS-GFP (stimulated). (B) iMEF[gt/gt] were co-transfected with expression plasmids for V5-tagged m152 together with either Cherry-STING or ev in combination with cGAS-GFP (stimulated) or ev (unstimulated). Twenty-four hours post-transfection, cells were fixed for immunolabeling with an anti-V5 antibody. White boxes indicate the region shown at a higher magnification. Scale bar represents 10 μm.

C  Lysates of Cherry-STING and either CD4-V5 or m152-V5 expressing 293T cells were subjected to immunoprecipitation (IP) with an anti-V5 antibody. Input and IP samples were analyzed by IB with the indicated antibodies.

D  iMEF stably expressing ev or m152-V5 were left unstimulated or stimulated with 10 μg/ml ISD and lysed 90 min later. m152 was immunoprecipitated with an anti-V5 antibody, and samples were analyzed by IB with V5, STING, and phospho-TBK1 (pTBK1)-specific antibodies.

Data information: IB shown are representative of three (C) or two (D) independent experiments.
Source data are available online for this figure.

perinuclear region (Fig 2B, second and third panel), consistent with our results in HeLa cells. Notably, upon co-transfection with cGAS and in the absence of STING, m152 remained in the ER and did not translocate to the perinuclear region (Fig 2B, last panel), suggesting that the translocation of m152 is dependent on STING. Since m152 and STING translocate simultaneously upon stimulation, we sought to examine whether these proteins interact with each other. Upon overexpression in 293T cells, STING co-immunoprecipitated with m152, but not with the control protein CD4 (Fig 2C). To show that m152 interacts with endogenous STING, we generated iMEF stably expressing V5-tagged m152 (Appendix Fig S2C) and either left them unstimulated or stimulated with ISD. Endogenous STING co-immunoprecipitated with m152 in unstimulated as well as stimulated cells (Fig 2D), correlating with our observation that m152 and STING co-localize regardless of stimulation status.

### The luminal N-terminal domain of m152 directs its interaction with STING

Next, we aimed to identify the domain of m152 which is essential for its interaction with STING and its effect on STING-mediated signaling. We constructed a series of chimeric m152 proteins by exchanging target domains singly or in combination with the corresponding region in CD4 (Fig 3A, (1)-(6)), a well-characterized cellular type I transmembrane protein which was previously used for the generation of chimeric proteins (Ziegler *et al*, 2000; Barton *et al*, 2006). All generated m152 mutants were expressed in 293T cells, and their localization was determined in unstimulated and stimulated conditions. Excepting the CD4-m152SPCTD (5) mutant, which only partially localizes with STING, wild-type m152 and all m152 mutants (1–4 and 6) co-localize with STING in unstimulated and stimulated cells (Fig EV1A). Next, we analyzed the m152-CD4 chimeric proteins in the established IFNβ luciferase reporter assay described previously. We observed that all m152 chimeras that still contained the luminal domain of m152 (1–4) retained the ability to inhibit signaling downstream of cGAS-STING (Fig 3B). Both m152-CD4 chimeric proteins where the N-terminal domain of m152 was replaced with that of CD4, namely CD4-m152SPCTD (5) and CD4-m152SPTMCTD (6), lost the ability to inhibit STING-mediated signaling (Fig 3B), showing that the N-terminal domain of m152 is responsible for its effect on STING signaling. Co-IP experiments in 293T cells confirmed that the N-terminal domain of m152 mediates its interaction with STING, whereas the transmembrane and C-terminal domain of m152 are inconsequential (Fig 3C). We also constructed an m152 mutant lacking N-linked glycosylation (Fink *et al*, 2013) (Fig 3A, (7)) and observed that post-translational N-linked glycosylation of m152 does not contribute to the impeding effect of m152 on STING-dependent signaling (Fig EV1B). In addition, we mutated the stalk region of m152, which is required for its binding to and effect on MHC class I (Janssen *et al*, 2016) (Fig 3A, (8)+(9)), and can show that this region is dispensable for its effect on STING signaling and interaction with STING (Fig 3D and E). These m152 mutants (7–9) also still co-localized with STING (Fig EV1A).

### The luminal loop regions of STING are the sites of interaction with m152

CMV and their respective hosts share a dynamic co-evolution spanning millions of years. While the m152 protein is highly conserved

among various MCMV strains, there is no known homologue in HCMV. Interestingly, we observed that m152 selectively inhibited IFNβ promoter activity downstream of murine STING (mSTING), but not human STING (hSTING) (Fig 4A).

To better understand why the effect of m152 is restricted to murine STING, we compared the protein sequences of murine and human STING and noted that, while the C-terminal domain of STING is conserved between both species, the predicted ER-luminal loop regions of STING are variable (Fig 4B). Since we have observed that the N-terminal domain of m152, which is localized in the ER lumen, is crucial for its interaction with STING, it raised the possibility that the loop regions of STING, which are likewise luminal, may be the reciprocal interaction site (Fig 4B).

To address this, we expressed various murine or human STING chimeras in the IFNβ luciferase assay to see which mutants would remain targets of m152. First, we introduced an N41E mutation into the first luminal loop region or replaced the second luminal loop region in murine STING with the respective sequences in human STING singly or in combination. Vice versa mutations were introduced in human STING (Fig 4B). The functionality of these STING mutants compared to WT murine or human STING was verified (Appendix Fig S3A).

Upon induction of the IFNβ promoter following overexpression of murine STING in which either the N41E mutation was introduced (mSTING-N41E) or the second loop region was exchanged (mSTING-hL2), m152 partially lost the ability to inhibit IFNβ promoter activity (Fig 4C). However, when both loops of murine STING were mutated (mSTING-N41E-hL2), m152 could no longer downmodulate signaling, just as m152 did not inhibit signaling downstream of human STING (Fig 4C). Co-immunoprecipitation experiments in 293T cells confirmed that these sites in STING also mediate its interaction with m152 (Fig 4D, lane 4; Appendix Fig S3B).

Similarly, signaling of human STING with introduced murine STING loop regions, namely hSTING-E41N and hSTING-mL2, was partially inhibited by m152, and hSTING-E41N-mL2 was inhibited by m152 similar to murine STING (Fig 4E). Likewise, while m152 interacted with hSTING only weakly (Fig 4F, lane 1), it co-precipitated with the human STING chimeras with murine STING loop regions (Fig 4F, lane 2–4; Appendix Fig S3B).

Altogether, these data reveal that m152 binds to both luminal loop regions of murine STING to exert its antagonistic effect on the STING-mediated type I IFN response.

### m152 antagonizes STING translocation, but not its activation or dimerization

Upon stimulation, STING follows a series of distinct steps prior to the activation of TBK1 and IRF3 and subsequent type I IFN transcription. Sequentially, these include the activation by 2′3′-cGAMP (i) and dimerization of STING (ii), its ER exit (iii), and translocation to the Golgi compartment (iv), where it activates downstream signaling via TBK1 and IRF3 (Fig 5A). We therefore sought to pinpoint which step m152 may target to inhibit STING-dependent signaling.

To assess whether m152 targets STING activation by cGAMP, we generated a ligand-independent, constitutively active mutant of STING by introducing a single aa substitution (V154M) (Jeremiah *et al*, 2014). As expected, expression of STING WT in the absence of cGAS did not lead to activation of the IFNβ promoter (Fig 5B). In comparison, expression of STING V154M induced activation of the

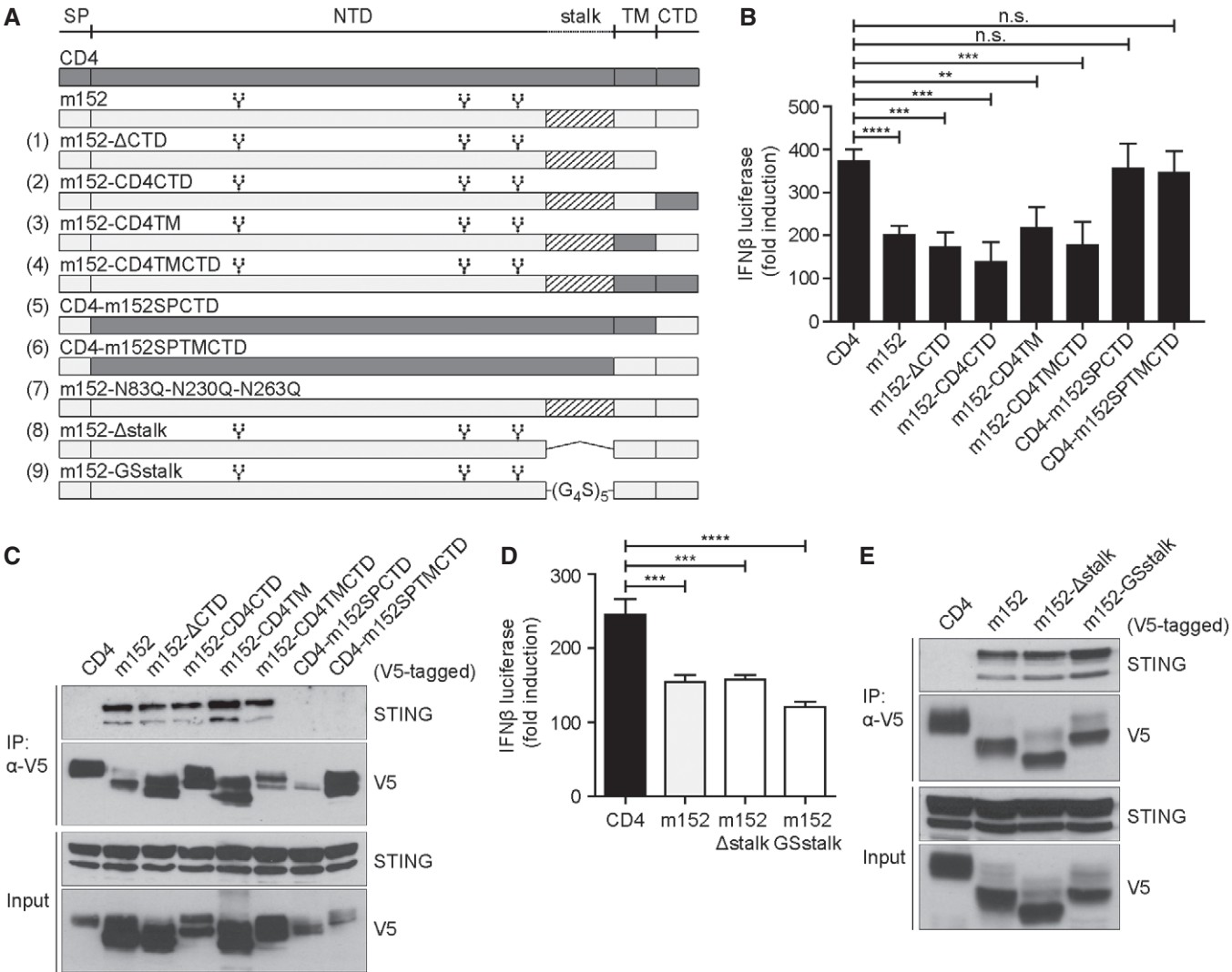

**Figure 3.  The N-terminal domain of m152 directs the interaction with STING.**

A   Schematic representation of wild-type m152, wild-type CD4, CD4-m152 chimeric constructs, m152 glycosylation mutants, and m152 stalk mutants used in this study. CD4-m152 chimeras (1–6): The relevant domain of m152 was replaced singly or in combination with the respective domain of murine CD4. m152 glycosylation mutant (7): asparagine (N) at position 83, 230, and 263 was mutated to glutamine (Q). m152 stalk mutants (8-9): m152-Δstalk has a deletion of amino acids 300-326 (8), and for m152-GSstalk, a glycine-serine linker ((G$_4$S)$_5$) was inserted to replace the stalk region (9). SP = signal peptide, NTD = N-terminal domain, TM = transmembrane domain, CTD = C-terminal domain. Branched symbols represent the three glycosylation sites of m152.

B   Cherry-STING, IFNβ-Luc, pRL-TK, cGAS-GFP (stimulated), or IRES-GFP (unstimulated) and indicated expression plasmids as shown in (A) were transiently expressed in 293T cells and a dual-luciferase assay was performed. Data are combined from two out of three independent experiments and shown as mean ± SD.

C   293T cells were co-transfected with expression plasmids for Cherry-STING and indicated chimeras as shown in (A). An anti-V5 IP was performed, and samples were analyzed by IB with indicated antibodies. IB shown is representative of three independent experiments.

D   Cherry-STING, IFNβ-Luc, pRL-TK, and either CD4, m152, m152-Δstalk, or m152-GSstalk were transiently expressed in 293T cells. For stimulation, samples were co-transfected with cGAS-GFP, and unstimulated samples with IRES-GFP. Lysates were analyzed as described in (B). Data are combined from two out of three independent experiments and shown as mean ± SD.

E   293T cells were co-transfected with expression plasmids for Cherry-STING and either CD4, m152, m152-Δstalk, or m152-GSstalk. An anti-V5 IP was performed, and samples were analyzed by IB with indicated antibodies. Immunoblot shown is representative of three independent experiments.

Data information: Student's *t*-test (unpaired, two-tailed), n.s. not significant, **$P < 0.01$, ***$P < 0.001$, ****$P < 0.0001$.
Source data are available online for this figure.

IFNβ promoter, confirming that this STING mutant can trigger downstream signaling independent of an activation step via cGAS. Nonetheless, in the presence of m152, signaling of STING V154M was still inhibited (Fig 5B), suggesting that m152 does not target STING activation by cGAMP, as observed before (Fig 1H).

To investigate whether m152 inhibits STING dimerization, we stimulated iMEF stably expressing m152 or corresponding control cells with ISD and assessed for the presence of STING dimers after stimulation (Fig 5C). We did not observe an effect in the kinetics of STING dimer formation when m152 was present (Fig 5C),

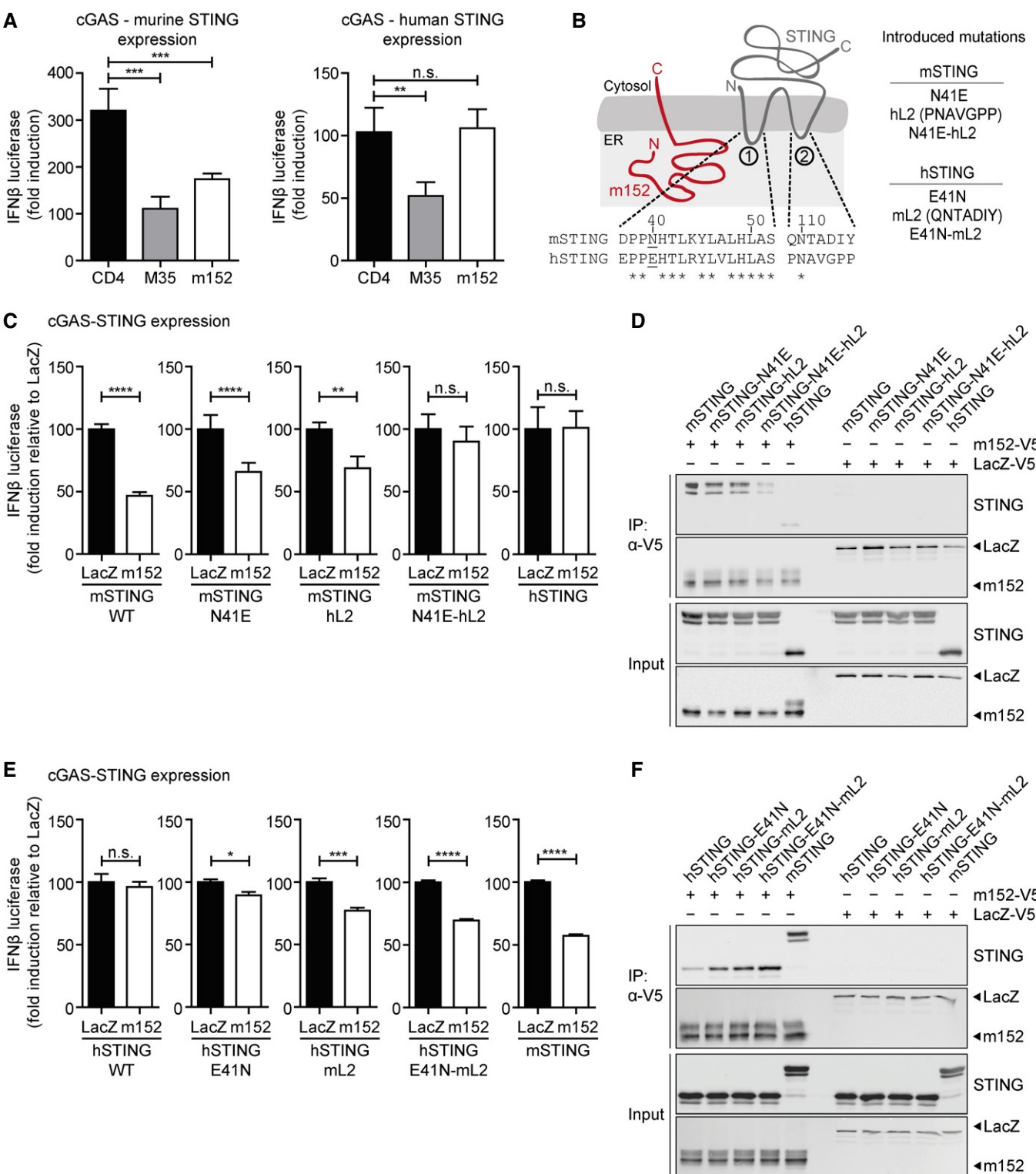

**Figure 4.**

which shows that m152 also does not affect the dimerization of STING.

Consistent with previous reports, we have observed that the ER-resident STING protein translocates to the perinuclear region following

activation (Fig 2A). To determine whether m152 affects the translocation kinetics of STING, we stimulated iMEF$^{gt/gt}$ Cherry-STING with ISD and monitored the translocation of STING by live cell imaging. As shown in Fig 5D, STING localizes in the ER in unstimulated cells and

**Figure 4.  Binding of m152 to both ER-luminal loop regions of murine STING is a prerequisite for its antagonistic activity.**

A    293T cells were co-transfected with expression plasmids for IFNβ-Luc, pRL-TK, cGAS-GFP (stimulated), or IRES-GFP (unstimulated) and indicated expression plasmids together with murine Cherry-STING (left panel) or human STING (right panel). Lysates were analyzed by dual-luciferase assay. Data are combined from two out of three independent experiments and shown as mean ± SD.

B    Schematic representation of the predicted m152 (red) and STING (gray) topology in the ER membrane. (1) and (2) specify the ER-luminal loop regions of STING with the sequence alignments from murine and human STING shown below. In murine STING, N41 in loop (1) was mutated to E41 and loop (2) was exchanged with loop (2) of human STING (hL2). The resultant constructs were designated mSTING N41E, mSTING hL2, and mSTING N41E-hL2. In human STING, mutations were introduced vice versa, resulting in hSTING E41N, hSTING mL2, and hSTING E41N-mL2.

C    IFNβ-Luc, pRL-TK, cGAS-GFP (stimulated), or IRES-GFP (unstimulated) and either LacZ or m152 together with the indicated murine STING mutants described in (B) were transiently expressed in 293T cells. Luciferase activity was measured as described in (A).

D    293T cells were co-transfected with expression plasmids for either LacZ or m152 together with the indicated murine STING mutants described in (B). An anti-V5 IP was performed, and samples were analyzed by IB with the indicated antibodies.

E, F  Luciferase assay (E) and co-IP (F) were performed as described in (C) and (D), respectively, using the indicated human STING mutants in place of the murine STING mutants described in (B).

Data information: Data in (C) and (E) are normalized to LacZ, combined from three independent experiments and shown as mean ± SD. IB shown in (D) and (F) are representative of three independent experiments. Student's *t*-test (unpaired, two-tailed), n.s. not significant, *P < 0.05, **P < 0.01, ***P < 0.001, ****P < 0.0001. Source data are available online for this figure.

upon stimulation, STING was present in clearly distinguishable perinuclear structures, which were used as a marker to quantify STING translocation. When we introduced m152 into these cells, we observed that the percentage of cells exhibiting translocated STING was markedly reduced compared to control cells (Fig 5E, representative images are shown in Fig EV2A). These data suggest that m152 delays the translocation of STING following its activation.

Upon stimulation of iMEF stably expressing m152 with ISD in the presence of Brefeldin A, which inhibits protein transport from the ER to the Golgi compartment, STING dimers were detected; however, phosphorylation of TBK1 did not occur (Fig EV2B). As already observed by others (Mukai *et al*, 2016), this suggests that STING dimerizes at the ER, whereas its translocation to the Golgi is a prerequisite for the activation of TBK1. Since STING translocation is important for the activation of TBK1 and therefore for the downstream activation of IRF3, we hypothesized that the hindrance of STING translocation by m152 would affect the phosphorylation of TBK1, IRF3, and STING itself. To test this hypothesis, we stimulated iMEF$^{gt/gt}$ stably expressing Cherry-STING and either m152 or the corresponding control cells with ISD and assayed for the phosphorylation of TBK1, IRF3, and STING in a time course (Fig 5F). In control cells, phosphorylation of TBK1 was already detected 30 min post-stimulation, while at the same time point in the presence of m152, TBK1 remained unmodified. Likewise, 90 min post-stimulation, phosphorylated IRF3 could be detected in control, but not in m152 expressing cells (Fig 5F). However, by 180 min post-ISD stimulation, phosphorylated TBK1 levels were comparable, regardless of the presence or absence of m152, while levels of phosphorylated IRF3 were still affected by the presence of m152, matching our observations from the live cell imaging experiments. Comparable results were also observed with endogenous STING in iMEF stably expressing m152 or the corresponding control cells (Fig EV2C). These data, in agreement with our live cell imaging results, indicate that m152 is not completely blocking STING-dependent signaling, but is rather delaying the translocation of STING to the perinuclear region and therefore delaying type I IFN induction.

## MCMV lacking m152 induces an elevated type I IFN response leading to control of viral transcription *in vitro*

So far, we have shown that ectopically expressed m152 inhibits the STING-dependent type I IFN response. Next, we wanted to assess whether m152 targets STING signaling in the context of MCMV infection. The Ly49H activating receptor on the cell surface of an NK cell subset from C57BL/6 mice plays a key role in early resistance to MCMV infection (Arase *et al*, 2002). To prevent interference or possible masking of the effect of m152 by highly activated NK cells in *in vivo* experiments, we conducted our studies with an MCMV mutant lacking the interaction partner of Ly49H, m157, hereinafter referred to as parental MCMV. On this background, we introduced a stop cassette in the m152 ORF to generate the recombinant MCMV m152stop (Fig 6A). We confirmed the intended mutagenesis as the m152 protein was only detected in iMEF upon infection with parental MCMV, but not MCMV m152stop, while expression of the immediate-early protein IE1 was comparable (Fig 6B). Additionally, we observed that the m152 protein is synthesized *de novo* very early during MCMV infection (Fig EV3A).

Since we have shown that ectopically expressed m152 interacts with STING (Fig 2D), we next examined whether this interaction is preserved during MCMV infection. We infected iMEF with parental MCMV or MCMV m152stop and iMEF$^{gt/gt}$ with parental MCMV alone and immunoprecipitated m152 (Fig 6C). Endogenous STING was detected in immunoprecipitates of parental MCMV-infected iMEF, but not in MCMV m152stop-infected iMEF (Fig 6C), demonstrating that m152 and STING interact during MCMV infection. To assess whether m152 inhibits type I IFN responses upon MCMV infection, we infected primary BMDM with parental MCMV or MCMV m152stop and observed that infection with MCMV m152stop led to higher levels of secreted IFNα and IFNβ compared to parental MCMV (Fig 6D). Additionally, via live cell imaging as described above, we observed that upon infection of iMEF$^{gt/gt}$ Cherry-STING with parental MCMV, which expresses m152, STING was translocated in significantly fewer cells compared to infection with MCMV m152stop, which does not express the m152 protein (Fig 6E, representative images are shown in Fig EV3B). In agreement with these results, phosphorylation of TBK1, IRF3, and STING is delayed upon infection of iMEFs with the parental MCMV compared to infection with MCMV m152stop (Fig EV3C).

Next, we sought to determine whether modulation of the type I IFN response by m152 impacts MCMV transcription. Since type I IFN levels are higher in the absence of m152, we would expect reduced transcription of viral genes. To test this, we infected iMEF with parental MCMV or MCMV m152stop and analyzed the

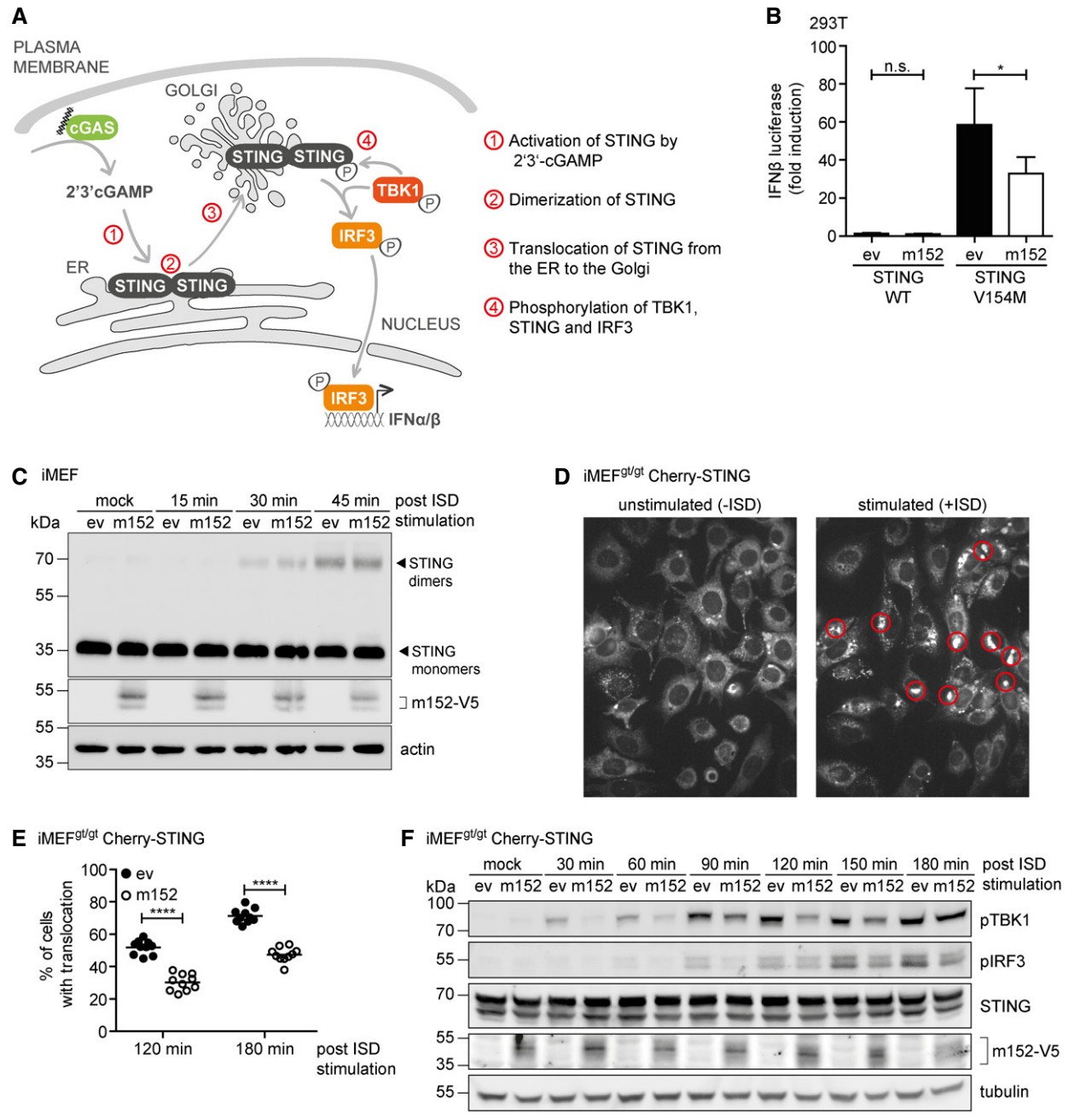

**Figure 5. m152 antagonizes STING translocation, but not its activation by cGAMP or its dimerization.**

A  Schematic representation of the key steps in the cGAS-STING signaling pathway.

B  293T cells were co-transfected with expression plasmids for IFNβ-Luc, pRL-TK, and either ev or m152. Cells were further co-transfected with either Cherry-STING WT or the constitutively active Cherry-STING V154M (stimulated) or with IRES-GFP (unstimulated). Data are combined from three independent experiments and shown as mean ± SD.

C  iMEF stably expressing ev or m152-V5 were stimulated with 10 μg/ml ISD for the indicated times or left unstimulated (mock). Cell lysates were analyzed by SDS–PAGE under non-reducing conditions and subjected to IB with the specified antibodies.

D  Two representative still images from live cell imaging experiments with iMEF^gt/gt stably expressing Cherry-STING transfected with ISD (right panel) or left unstimulated (left panel). Red circles highlight representative translocated STING in ISD stimulated cells, which is used as an indicator of activation.

E  iMEF^gt/gt stably expressing Cherry-STING and either ev or m152-V5 were stimulated with ISD. Live cell imaging was performed and STING translocation quantified 120 and 180 min post-stimulation. Data shown are one representative of two independent experiments.

F  iMEF^gt/gt stably expressing Cherry-STING and V5-tagged m152 or corresponding ev were stimulated with 5 μg/ml ISD for the indicated time or left unstimulated (mock). Lysates were subjected to IB with the specified antibodies.

Data information: IB shown in (C) and (F) are representative of three and two independent experiments, respectively. Student's *t*-test (unpaired, two-tailed), n.s. not significant, *$P < 0.05$, ****$P < 0.0001$.

Source data are available online for this figure.

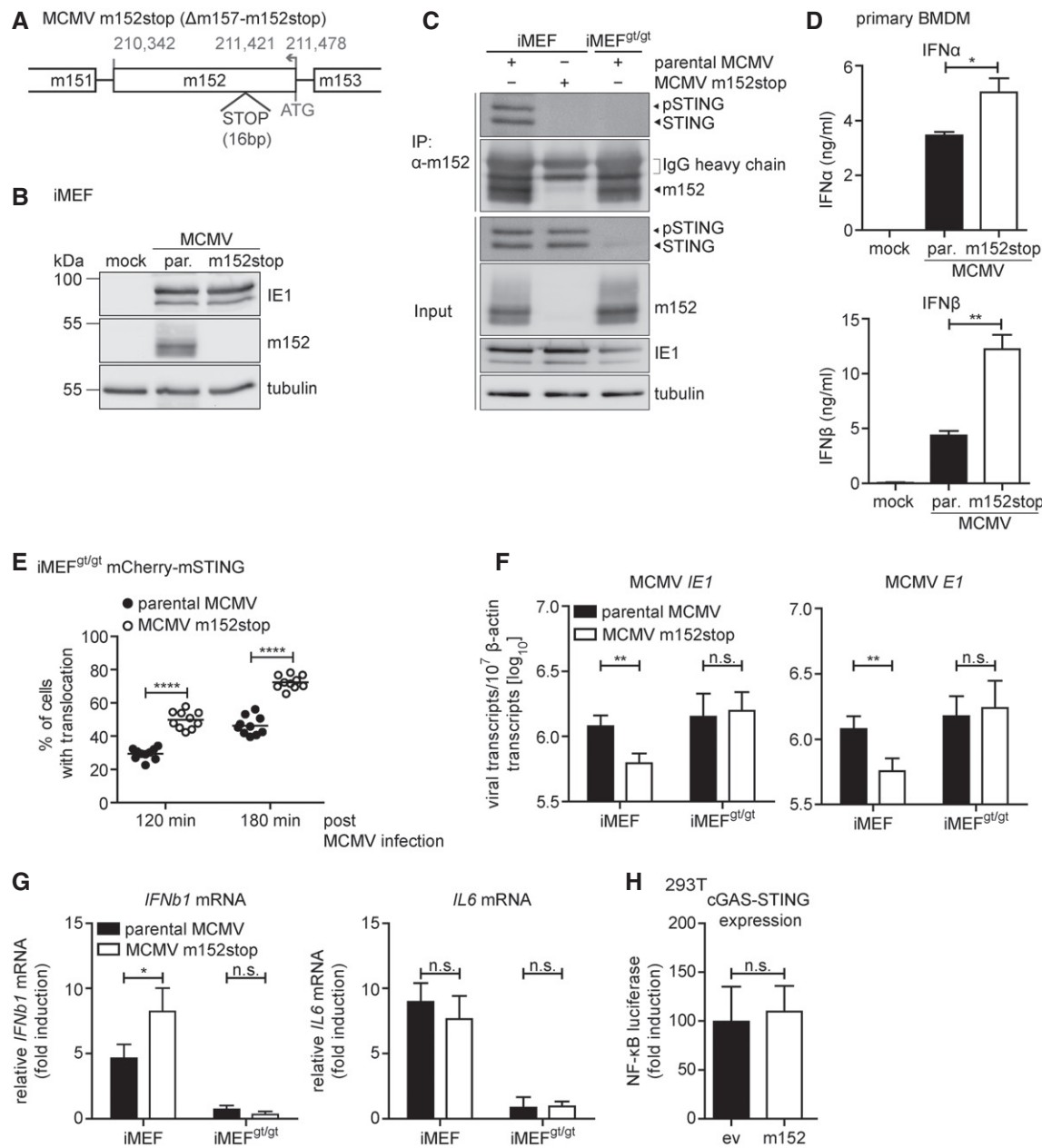

**Figure 6.  MCMV lacking m152 induces an elevated type I IFN response leading to lower levels of viral transcripts *in vitro*.**

A    Schematic representation of the recombinant MCMV m152stop virus constructed on the MCMV Δm157 backbone (here referred to as parental MCMV and MCMV m152stop for simplicity). Shown is the transcriptional coding region of m152. STOP indicates the introduced 16 base pair (bp) stop cassette.

B    iMEF were infected by centrifugal enhancement with either parental MCMV (par.) or MCMV m152stop at an MOI of 0.5 or mock infected. Three hours post-infection (hpi), cells were lysed and subjected to immunoblotting (IB) with the specified antibodies.

C    iMEF and iMEF^gt/gt were infected by centrifugal enhancement with parental MCMV or MCMV m152stop, or parental MCMV alone, respectively (MOI 0.5). Three hpi lysates were subjected to an anti-m152 IP, and samples were analyzed by IB with indicated antibodies. IB shown is representative of two independent experiments.

D    Primary BMDM were infected with parental MCMV (par.) or MCMV m152stop at an MOI of 0.1 or mock infected. At 16 hpi, secreted levels of IFNα or IFNβ were quantified by ELISA. Data are representative of two independent experiments.

E    iMEF^gt/gt stably expressing Cherry-STING were infected with parental MCMV or MCMV m152stop (MOI 0.5). Live cell imaging was performed and STING translocation quantified 120 and 180 min post-stimulation. Data shown are representative of two independent experiments.

F, G  iMEF or iMEF^gt/gt were infected by centrifugal enhancement with parental MCMV or MCMV m152stop (MOI 0.01). At 6 hpi, total RNA was extracted to determine MCMV *IE1* and MCMV *E1* (F), *IFNb1* and *IL6* (G) transcripts by qRT–PCR. Data shown are combined from two out of three independent experiments.

H    293T cells were co-transfected with Cherry-STING, the pNF-κB luciferase reporter, pRL-TK, cGAS-GFP (stimulated), or IRES-GFP (unstimulated) and either ev or m152. Cells were lysed and analyzed as described in Fig 1. Data are combined from three independent experiments.

Data information: Student's *t*-test (unpaired, two-tailed), n.s. not significant, *$P < 0.05$, **$P < 0.01$. Data are shown as mean ± SD.
Source data are available online for this figure.

transcript levels of MCMV genes *IE1* and *E1* 6 hours post-infection (hpi) (Fig 6F). In the absence of m152, reduced *IE1* and *E1* transcript levels were detected, indicating that m152-mediated inhibition of STING is required for efficient viral transcription at this early stage of MCMV infection. As a control, m152 transcripts in parental MCMV-infected cells were present at comparable levels 6 hpi in STING-proficient and STING-deficient cells (Fig EV3D). To show that MCMV transcription is affected by m152-mediated inhibition of STING-dependent IFN signaling, we included STING-deficient MEFs, iMEF$^{gt/gt}$ in this experiment. In iMEF$^{gt/gt}$, *IE1* and *E1* transcript levels were identical upon both parental MCMV and MCMV m152stop infection (Fig 6F), demonstrating that the effect on MCMV transcription exerted by m152 is ameliorated in the absence of STING. Unexpectedly, we observed that viral transcript levels were not elevated in iMEF$^{gt/gt}$ (Fig 6F) as it would be expected if STING had a solely antiviral role. Next, we examined cytokine levels by measuring *IFNb1* and *IL6* mRNA transcript levels in iMEF and iMEF$^{gt/gt}$ infected with parental MCMV or MCMV m152stop (Fig 6G). As observed in iBMDM, *IFNb1* mRNA levels were elevated in iMEF infected with MCMV m152stop, and as expected, no induction of *IFNb1* was detectable in the absence of STING (Fig 6G). Additionally, *IL6* mRNA induction, which is mediated by NF-κB, was completely dependent on STING (Fig 6G). This result may shed a light on our observation that the absence of STING did not elevate viral transcript levels (Fig 6F), since it has been shown that NF-κB signaling is crucial for early MCMV replication (Isern *et al*, 2011). Notably, *IL6* mRNA levels in iMEF were not affected by m152, which suggests that m152 specifically targets STING-dependent IRF3, but not NF-κB signaling. Indeed, m152 did not affect activation of NF-κB-mediated transcription upon cGAS-STING expression in a luciferase reporter assay (Fig 6H).

So far, our experiments have been conducted in cells derived from B6J mice. Previously, m152 has been described to prevent cell surface expression of the NKG2D ligands RAE-1α/β/γ (Lodoen *et al*, 2003), which are expressed in Balb/c, but not in B6J mice. To exclude competition between STING and these NKG2D ligands, we infected primary MEFs from B6J and Balb/c mice in parallel with wild-type MCMV (MW97.01) (Wagner *et al*, 1999) or the previously characterized m152 deletion mutant MCMV Δm152 (Wagner *et al*, 2002), and assessed cytokine expression. As in B6J MEFs, we observed that m152 inhibits *IFNb1*, but not *IL6* transcription in Balb/c MEFs (Fig EV4A).

Taken together, these results show that m152 inhibits the STING-mediated type I IFN response upon MCMV infection *in vitro*, leading to reduced type I IFN secretion and increased viral transcription, while STING-dependent NF-κB activation and IL-6 induction are not affected by m152.

### Reduced transcription of MCMV lacking m152 *in vivo* is mediated by STING-dependent signaling

We next addressed the question if the inhibitory effect of m152 on the type I IFN response and subsequent viral transcription is preserved *in vivo*. B6J or STING$^{-/-}$ mice were infected with parental MCMV or MCMV m152stop and type I IFN levels were analyzed in spleen homogenates and serum 6 hpi (Fig 7A and B). We observed significantly elevated IFNα and IFNβ levels in the spleen and serum of B6J mice infected with MCMV m152stop compared to infection

with parental MCMV (Fig 7A and B). Consistent with previous observations that STING-dependent signaling is crucial for the induction of the initial type I IFN response against MCMV infection (Lio *et al*, 2016), IFNα was barely detectable in STING$^{-/-}$ mice, and IFNβ levels were below the detection limit (Fig 7A and B). Remarkably, the levels of the NF-κB-induced proinflammatory cytokine IL-6 were not affected by the presence of m152 in B6J mice (Fig 7C) as observed *in vitro* (Fig 6G). STING$^{-/-}$ mice did not mount an IL-6 response (Fig 7C), as observed previously in *in vitro* settings (Fig 6G), showing that STING is essential for this early IL-6 response to MCMV infection. These results confirm our *in vitro* findings that m152 selectively inhibits IRF3-mediated activation of type I IFN transcription, but not NF-κB-mediated IL-6 transcription.

To examine whether the inhibition of type I IFN signaling by m152 is crucial for MCMV infection in its host, we examined viral transcript levels upon infection with parental MCMV or MCMV m152stop. Consistent with our observations *in vitro* (Fig 6F), MCMV *IE1* and *E1* transcripts were significantly reduced in B6J mice infected with MCMV m152stop compared to infection with parental MCMV 6 hpi (Fig 7D). Importantly, this suppression in MCMV transcription was rescued in mice lacking STING (Fig 7D), showing that decreased viral transcription in the absence of m152 is mediated by STING-dependent signaling. In addition, when we infected Balb/c mice with wild-type MCMV (MW97.01) or Δm152 MCMV, we observed similar trends for IFNα and IL-6 levels (Fig EV4B) as well as MCMV *IE1* and *E1* transcription (Fig EV4C) as for parental MCMV and MCMV m152stop in B6J mice (Fig 7A–D).

To address the possibility of cross interference between the previously characterized inhibitory effect of m152 on very early NK cell responses and the STING-mediated type I IFN response, we depleted B6J or STING$^{-/-}$ mice of NK cells prior to infection with parental MCMV or MCMV m152stop for assessment of MCMV transcription (Fig 7F). As we observed comparable results in the presence or absence of NK cells, we conclude that the effect of m152 on STING-dependent signaling is independent of its effect on the NK cell responses. Notably, similar to our observations in MCMV-infected iMEF (Fig 6F), MCMV *IE1*, *E1,* and *m152* transcript levels of parental MCMV and MCMV m152stop were comparable or even reduced in STING$^{-/-}$ mice (Fig 7D–F). This reveals a potentially dual role for STING, whereby it serves to both restrict and promote MCMV infection *in vivo*.

### STING activates NF-κB signaling from the ER

Taken together, our results have so far shown that m152 delays the translocation of STING upon stimulation, thereby inhibiting the type I IFN response *in vitro* and *in vivo*. Interestingly, MCMV transcript levels were not elevated at early stages of infection in the absence of STING (Figs 6F and 7D), as would be expected if STING acts solely as a viral restriction factor. This raised the question of whether STING's role during MCMV infection is exclusively antiviral. Moreover, we observed that the NF-κB response remains intact in the presence of m152 (Fig 6H). We therefore hypothesized that STING may activate NF-κB signaling before translocating from the ER to the Golgi compartment and that this STING-dependent NF-κB response may be beneficial for early MCMV transcription. To test this, we screened for a STING mutant which remains in the ER upon stimulation, and consequently does not induce a

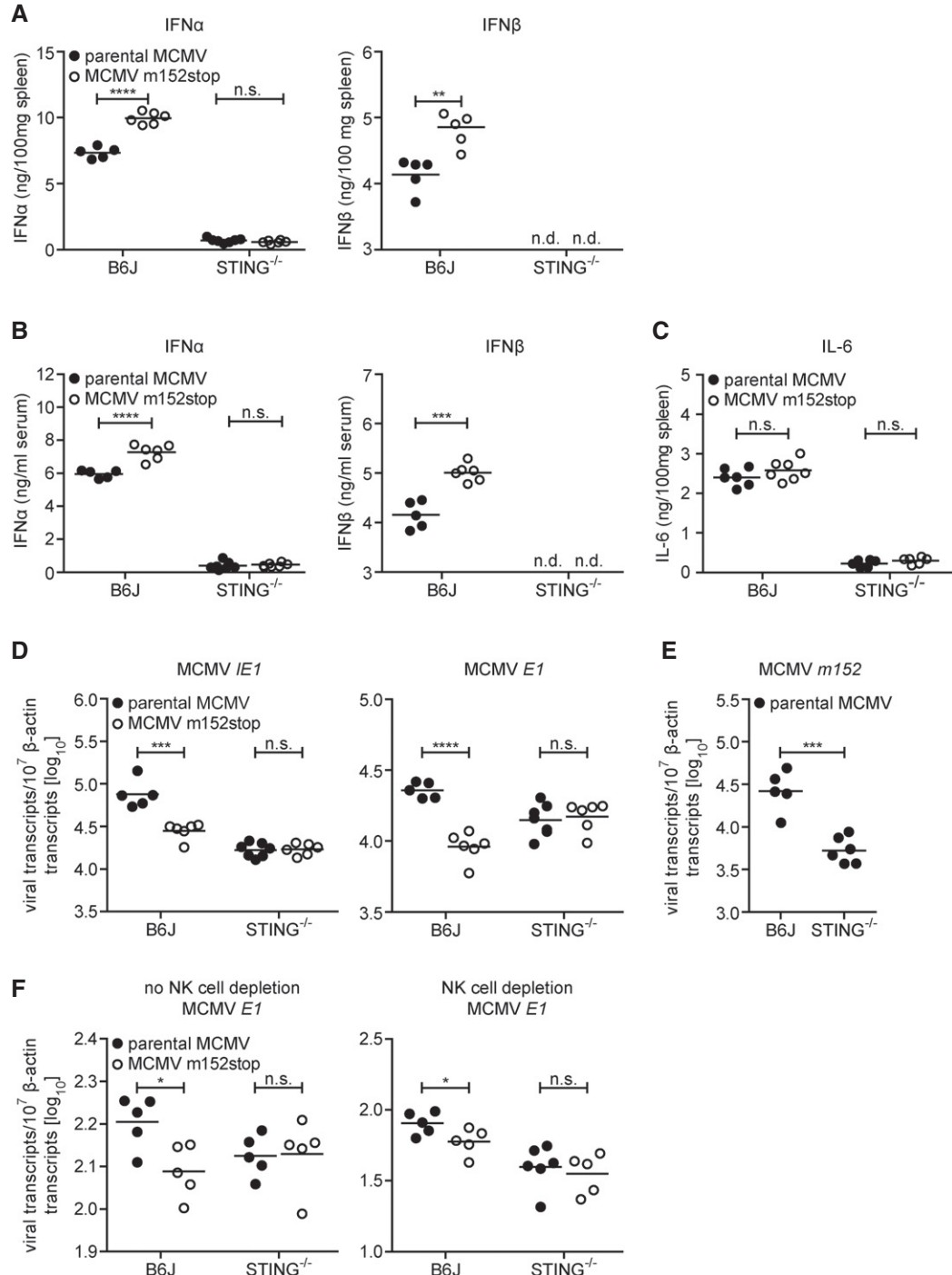

**Figure 7.   In the absence of STING, the impairment of MCMV m152stop *in vivo* is rescued.**

A–C   Type I IFN levels in spleen homogenates (A) and serum (B) and IL-6 levels in spleen homogenates (C) of C57BL/6J (B6J) or STING$^{-/-}$ mice following i.v. infection with 4 × 10$^5$ PFU parental MCMV or MCMV m152stop were analyzed 6 hpi by ELISA. IFNβ levels of STING$^{-/-}$ mice were below the detection limit (n.d.).

D   B6J or STING$^{-/-}$ mice were i.v. infected with 4 × 10$^5$ PFU parental MCMV or MCMV m152stop. Six hpi, RNA was extracted from spleen homogenates and expression of MCMV *IE1* and *E1* transcripts was determined by qRT–PCR.

E   B6J or STING$^{-/-}$ mice were i.v. infected with 4 × 10$^5$ PFU parental MCMV. Six hpi, RNA was extracted from spleen homogenates and *m152* transcript levels were determined by qRT–PCR.

F   B6J or STING$^{-/-}$ mice were depleted of NK cells via treatment with anti-NK1.1 (right panel) or left untreated (left panel). One day after NK cell depletion, B6J or STING$^{-/-}$ mice were i.v. infected with 4 × 10$^5$ PFU parental MCMV or MCMV m152stop. Sixteen hpi, RNA was extracted from spleen homogenates and MCMV *E1* transcript levels were determined by qRT–PCR. Data were normalized to 10$^7$ cellular β-actin transcripts (D–F).

Data information: *n* = 5–6 mice per group; Student's *t*-test (unpaired, two-tailed), n.s. not significant, **P < 0.01, ***P < 0.001, ****P < 0.0001.

type I IFN response. For human STING, it was previously shown that ubiquitination is a prerequisite for STING translocation (Ni *et al*, 2017). We therefore mutated several lysine residues to arginine in the cytosolic domain of murine STING and performed a cGAS-STING luciferase assay using luciferase reporters responding to IRF- and NF-κB signaling (p125 luciferase), to IRF signaling only (p125AA luciferase), or to NF-κB signaling only (NF-κB luciferase) (Appendix Fig S4). Most mutations (K83R, K235R, K275R, K337R) did not alter the capacity of STING to induce IRF- and NF-κB signaling. In contrast, the K288R STING mutant showed strongly impaired activation of both IRF-responsive reporters (p55-CIB luciferase and p125AA luciferase), while it was only slightly impaired in its capability to activate NF-κB-dependent reporters (NF-κB luciferase and p125 luciferase) (Fig 8A). When expressed in 293T cells, the K288R STING mutant remained in the ER upon stimulation, while WT STING translocated to the perinuclear region (Fig 8B). These results show that despite remaining in the ER upon stimulation, the K288R STING mutant can still mount an NF-κB response. To verify this, we transduced iMEF$^{gt/gt}$ stably expressing m152-V5 or corresponding control cells with either Cherry-tagged WT STING or K288R STING and assessed cytokine expression. As shown before, upon stimulation with ISD, WT STING induced the transcription of *IFNb1* and *IL6* mRNA, and the presence of m152 inhibited the induction of *IFNb1*, but not *IL6* mRNA (Fig 8C and D). In agreement with our luciferase reporter and IF assays, STING K288R could not induce the transcription of *IFNb1* mRNA, however, induction of *IL6* transcription was preserved (Fig 8C and D). In addition, iMEF$^{gt/gt}$ stably expressing Cherry-tagged K288R STING induced NF-κB (p65) nuclear translocation upon MCMV infection as well as iMEF$^{gt/gt}$ stably expressing Cherry-tagged WT STING, while no p65 nuclear translocation was observed in iMEF$^{gt/gt}$ lacking STING expression (Fig EV5). As shown in Fig EV2B, STING translocation is crucial to activate the kinase TBK1. To evaluate whether TBK1 is needed for STING-dependent NF-κB activation, we performed an siRNA-mediated knockdown of TBK1 in 293T cells (Appendix Fig S5A) and performed a cGAS-STING luciferase assay with IRF and NF-κB response reporter constructs. While WT STING and K288R STING did not activate the IRF-responsive p55-CIB luciferase reporter in TBK1 knockdown cells, activation of the NF-κB response luciferase reporter was similar in TBK1-proficient and TBK1-deficient cells (Appendix Fig S5B). Taken together, these results show that STING activates NF-κB-dependent signaling at the ER independent of the kinase TBK1.

### STING-mediated NF-κB activation promotes early MCMV transcription

With the K288R STING mutant, we could now distinctly dissect the NF-κB and IRF arms of STING signaling as well as test our hypothesis that STING-mediated NF-κB activation promotes MCMV transcription. While MCMV infection of cells expressing WT STING will mount both an NF-κB and IRF response, infection of K288R STING expressing cells only results in an NF-κB response. We infected iMEF$^{gt/gt}$, iMEF$^{gt/gt}$ stably expressing Cherry-tagged WT STING, or K288R STING with the parental MCMV and MCMV m152stop and analyzed viral transcript levels (Fig 8E). As shown earlier (Fig 6F), MCMV *E1* transcript levels were not affected by the presence of m152 in iMEF$^{gt/gt}$, while infection of WT STING expressing cells

with MCMV m152stop led to reduced MCMV *E1* transcript levels compared to infection with parental MCMV (Fig 8E). In agreement with our findings that m152 targets STING trafficking, the presence of m152 did not affect viral transcription in cells expressing the K288R STING mutant, since this mutant does not translocate upon activation (Fig 8E).

Notably, in the presence of K288R STING, which only activates NF-κB, but not IRF, signaling viral transcript levels upon infection with MCMV m152stop were significantly higher compared to iMEF$^{gt/gt}$, in which the type I IFN response is completely abrogated, and in iMEF$^{gt/gt}$ stably expressing WT STING, which can mount an IRF response unaffected by m152 (Fig 8E).

Moreover, we observed that the viral transcript levels upon infection with the parental MCMV are significantly higher in cells stably expressing either WT STING or K288R STING compared to iMEF$^{gt/gt}$ (Fig 8E). This shows that parental MCMV also benefits from the presence of STING, which is contrary to the current paradigm of STING acting solely as a restriction factor for MCMV.

As mentioned before, both K288R STING and WT STING activate NF-κB signaling, whereas only WT STING additionally induces the antiviral type I IFN response. Our data clearly suggest that MCMV evolved the m152 protein to modulate the STING-mediated type I IFN response, while leaving the NF-κB response untouched. Hence, viral transcript levels in WT STING and K288R STING expressing cells are expected to be comparable in the presence of m152. Indeed, when we infected WT or K288R STING expressing cells with parental MCMV (which expresses m152), we do not observe a significant increase in viral transcript levels. This once again shows that MCMV profits from intact STING-mediated NF-κB signaling.

Taken together, our study identifies MCMV m152 as a novel viral antagonist of the STING-mediated antiviral type I IFN response and reveals that STING activates NF-κB signaling already from the ER prior to trafficking to the Golgi which is beneficial for early MCMV transcription.

## Discussion

The cGAS-STING signaling pathway plays a pivotal role in the antiviral innate immune response with the number of corresponding viral antagonists rising steadily. While some herpesviral antagonists target cGAS directly (Wu *et al*, 2015; Zhang *et al*, 2016; Su & Zheng, 2017), others mediate the degradation of STING (Kim *et al*, 2017) or target downstream signaling pathways (Christensen *et al*, 2016). However, the m152 protein is a clear stand out: It modulates this pathway differentially by antagonizing cGAS-STING-mediated activation of type I IFN signaling, while leaving cGAS-STING-mediated activation of NF-κB signaling intact, and it does so by delaying trafficking of STING from the ER to the Golgi compartment (Fig 9). Through our detailed mechanistic study of this novel and selective STING modulator, we have uncovered that STING activates NF-κB signaling already from the ER, prior to its trafficking (Fig 9). Furthermore, our study is the first to show the necessity of STING-mediated NF-κB signaling for CMV transcription *in vitro*, as well as the necessity of modulation of the STING-mediated type I IFN response for CMV replication *in vitro* and *in vivo*.

We have shown that the m152 protein co-localizes and interacts with STING in unstimulated cells in the ER and likewise traffics with

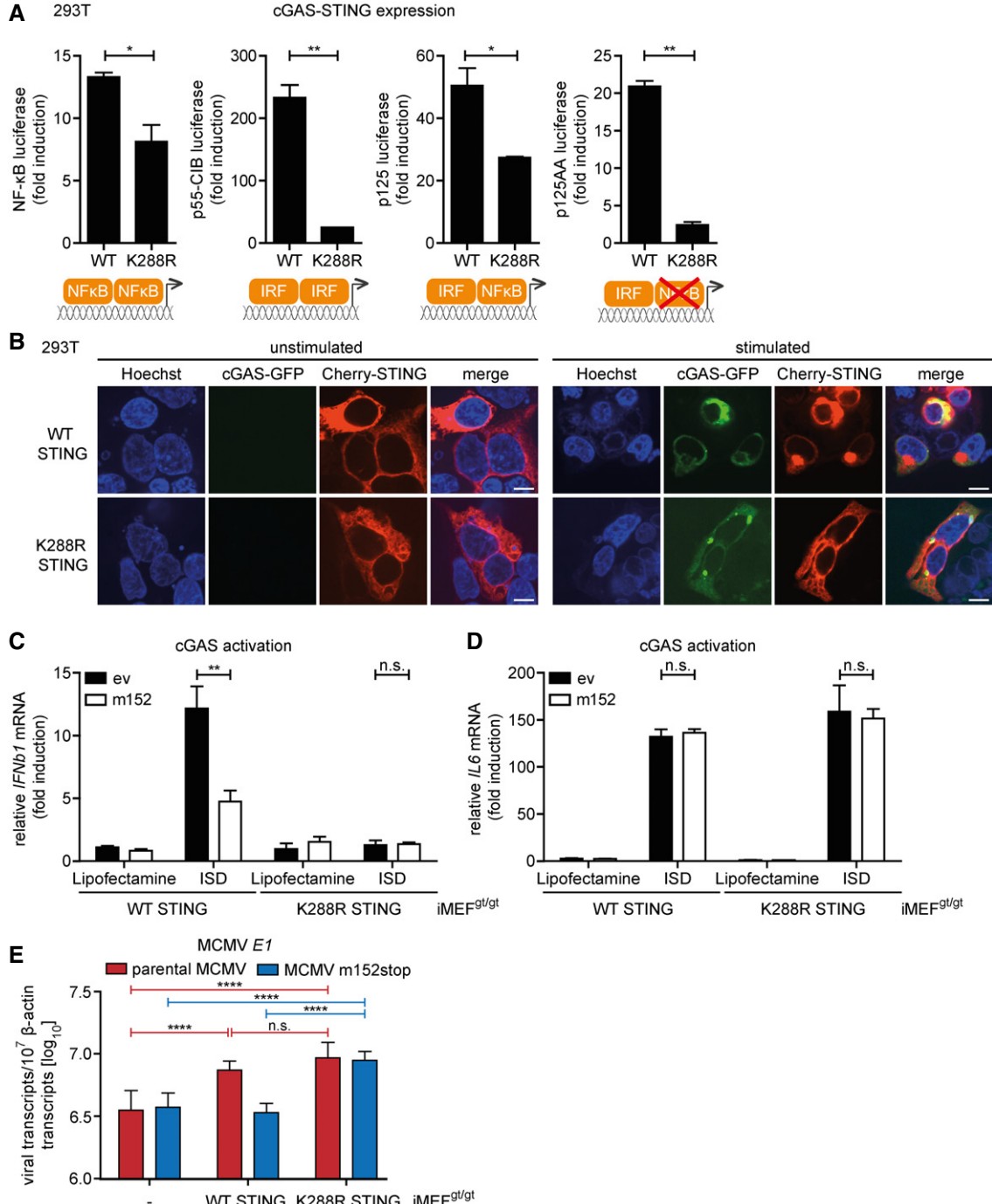

**Figure 8.   The STING-mediated NF-κB response is activated from the ER and specifically pro-viral for MCMV transcription.**

A   293T cells were co-transfected with Cherry-STING, pRL-TK, cGAS-GFP (stimulated), or IRES-GFP (unstimulated) and either the pNF-κB, p55-CIB, p125, or p125AA luciferase reporter. Data are representative of two independent experiments.

B   293T cells were co-transfected with expression plasmids for cGAS-GFP (stimulated) or ev (unstimulated) together with either cherry-tagged WT STING or cherry-tagged K288R STING. Cells were fixed for imaging 24 h post-transfection. Scale bar represents 10 μm.

C, D   iMEF^gt/gt stably expressing cherry-tagged WT STING or K288R STING and either ev or V5-tagged m152 were stimulated with ISD (10 μg/ml) or mock stimulated. At 4 hpi, total RNA was extracted to determine *IFNb1* (C) and *IL6* (D) transcripts by qRT–PCR. Data shown are combined from three (C) or two (D) out of three independent experiments.

E   iMEF^gt/gt (-) and iMEF^gt/gt stably expressing either cherry-tagged WT STING or K288R STING were infected by centrifugal enhancement with parental MCMV or MCMV m152stop (MOI 0.01). Six hpi total RNA was extracted to determine MCMV *E1* transcripts by qRT–PCR. Data shown are combined from three independent experiments.

Data information: Student's *t*-test (unpaired, two-tailed), n.s. not significant, *$P < 0.05$, **$P < 0.01$, ****$P < 0.0001$. Data are shown as mean ± SD.

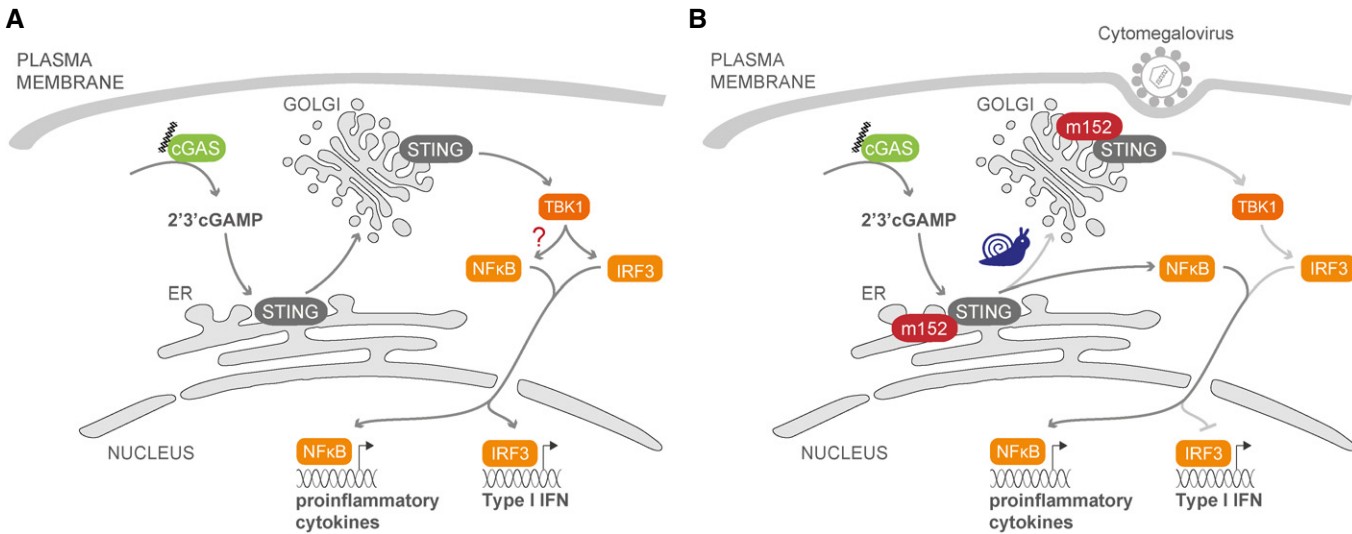

**Figure 9. The multifaceted m152 protein selectively modulates STING-dependent type I IFN, but not NF-κB, signaling.**

A   Upon stimulation with DNA or by viral infection, the pattern recognition receptor (PRR) cGAS produces the second messenger 2′3′-cGAMP, which binds to the ER-resident protein STING. STING then dimerizes and translocates from the ER to the Golgi, from where it activates the signaling pathway leading to induction of type I IFN via the kinase TBK1 and the IRF3 transcription factor. cGAS-STING signaling also induces activation of the NF-κB transcription factor, leading to proinflammatory cytokine expression, but from which subcellular compartment this signaling pathway is initiated is not understood.

B   The viral type I membrane protein m152 is expressed immediately after MCMV infection. m152 binds STING in the ER via their respective luminal domains and traffics with STING to the Golgi. In the presence of m152, trafficking of STING is delayed, leading to a reduced type I IFN, but intact NF-κB, response. Mutation of K288 in STING results in a STING mutant that cannot leave the ER and thereby cannot activate the type I IFN response, but is still able to induce the NF-κB pathway. These results suggest that STING induces the NF-κB pathway from the ER, prior to its trafficking to the Golgi.

STING to the Golgi compartment in stimulated cells (Fig 2). Exemplifying the strict host specificity and the host adaptation of CMVs, m152 interacts with murine, but not human STING. Using targeted chimeras, we could show that this interaction is directed by the N-terminal domain of m152 and the two ER-luminal loop regions of STING (Figs 3 and 4). Upon dissection of the STING signaling pathway, we show that m152 has no effect on STING activation by cGAMP or on STING dimerization, but delays its translocation from the ER to the Golgi compartment, which we show by ectopic expression of m152 and in the context of MCMV infection (Fig 5). The delay in STING trafficking mediated by m152 specifically resulted in inhibition of the type I IFN response, which is initiated by STING via TBK1 and IRF3 from the Golgi compartment and requires STING translocation. On the contrary, the STING-dependent NF-κB response was not inhibited by m152. While infection with an MCMV lacking m152 induced elevated type I IFN responses both *in vitro* and *in vivo*, leading to tighter host control of viral transcription at early time points post-infection, the NF-κB-dependent IL-6 response was not affected by m152 (Figs 6 and 7). Taken together, these results suggested that murine STING may activate the NF-κB pathway already at an earlier stage, probably at the ER membrane, prior to its trafficking to the Golgi compartment (Fig 9). Based on this hypothesis, we generated and characterized the STING mutant K288R, which remains ER-bound and can therefore not induce type I IFN signaling. We could show that the STING K288R mutant was indeed still able to induce an NF-κB response similar to WT STING (Fig 8). This finding is in line with a previous study showing that the human STING mutant K224R is translocation-deficient, but can still trigger NF-κB signaling upon stimulation (Ni *et al*, 2017). However, the corresponding mutation of

K288R in human STING, K289R, was reported to lead to a hyperactive version of human STING (Ni *et al*, 2017).

Interestingly, although we expected strongly enhanced MCMV transcription in the absence of STING, since it is a major host restriction factor of MCMV, we observed slightly reduced viral transcript levels in STING$^{-/-}$ mice (Fig 7). Notably, we observed that NF-κB-dependent IL-6 induction, as well as IRF3-dependent type I IFN induction, is completely dependent on STING *in vitro* and *in vivo* at this early stage of MCMV infection (Figs 6 and 7). Studies with the HCMV major immediately early promoter (MIEP) enhancer have shown that NF-κB interaction with corresponding binding sites in the enhancer element is essential for MIEP activity and initiation of the viral transcriptional program (Caposio *et al*, 2007). In addition, it has been shown that a combinatorial knockout of NF-κB and AP-1 sites in the MCMV MIEP enhancer leads to reduced MCMV replication (Isern *et al*, 2011). To show whether STING-mediated activation of NF-κB is indeed required for initial MCMV replication, we complemented STING-deficient murine fibroblasts with the STING K288R mutant and found enhanced MCMV transcript levels compared to cells lacking STING signaling (Fig 8), strongly suggesting that the STING-mediated NF-κB response is specifically proviral. Hence, our findings reveal a dual role for STING, whereby it serves to both restrict and promote MCMV infection *in vivo*.

The m152 protein has an impressive immune evasion repertoire. m152 has previously been described to evade NK cell- and T cell-dependent immune responses (Krmpotic *et al*, 2002; Fink *et al*, 2013). m152 prevents cell surface expression of the NKG2D ligands RAE-1α/β/γ/δ/ε (Lodoen *et al*, 2003), and for RAE-1γ and RAE-1ε, it was demonstrated that m152 retains these molecules in the

ERGIC, presumably by direct interaction (Arapovic *et al*, 2009a,b; Wang *et al*, 2012). In addition, m152 was described as a modulator of antigen presentation to CD8 T cells (Ziegler *et al*, 1997; Krmpotic *et al*, 1999). m152 retains peptide-loaded MHC class I molecules in the ERGIC, thereby preventing MHC class I surface expression (Ziegler *et al*, 1997; Lemmermann *et al*, 2010). These findings raise the question of whether the effect of m152 on STING, MHC class I, and RAE-1 proteins may be linked with each other. The m152-mediated delay in STING trafficking is most likely independent of its interaction with MHC class I, since we show that the m152 stalk region is dispensable for its interaction with STING and downmodulation of STING-mediated type I IFN signaling (Fig 3), whereas the stalk region is required for the interaction between m152 and MHC class I (Janssen *et al*, 2016). In addition, the effect m152 exerts on the STING-mediated type I IFN response occurs at a very early stage of MCMV infection, while the CD8 T cell response follows days later. Our NK cell depletion experiments clearly show that the reduced transcriptional activity of MCMV m152stop at 16 hpi is not attributed to the NK cell response, nor is it a result of the inhibitory effect of m152 on NK cell activation, but is solely due to the interaction between m152 and STING (Fig 7). In addition, the RAE-1 ligands that are downregulated most efficiently by m152, RAE-1α/β/γ (Lodoen *et al*, 2003), are not present in B6J mice (Lodoen *et al*, 2003; Arapovic *et al*, 2009b), in which most of our experiments have been conducted. This together with our NK cell depletion data *in vivo* clearly shows that m152 modulates the STING-mediated type I IFN response independently of its effect on RAE-1.

It is fascinating how many strategies MCMV has evolved to inhibit PRR signaling: The M27 protein inhibits signaling downstream of the IFNAR (Zimmermann *et al*, 2005). Notably, the tegument protein M45 first activates NF-κB signaling immediately upon MCMV infection (Krause *et al*, 2014), whereas *de novo* expressed M45, which is expressed later than m152, shuts down the proinflammatory cytokine response downstream of TLR signaling (Fliss *et al*, 2012). We have shown that the tegument protein M35 greatly reduces NF-κB-mediated transcription of IFNβ downstream of multiple PRR (Chan *et al*, 2017). This shows how precisely MCMV modulates the innate immune response, always adjusted to its needs and according to the stage of viral replication. We have now expanded on the scope and intricacies of the immune modulation mechanisms employed by MCMV to subvert multiple facets of the host immune defense via a single multifunctional immune evasion protein. In addition, the delay in STING-dependent type I IFN signaling, but not NF-κB signaling, provides a crucial advantage for MCMV in the establishment of an infection. In addition, m152 has provided us with novel insights into the precise steps of the crucial STING signaling pathway. By dissecting the immune evasion strategies exerted by MCMV m152, it becomes apparent that CMV has evolved elegant ways to not only modulate immune detection, but also actively manipulate and balance host responses to create an optimal environment for infection.

# Materials and Methods

## Ethics statement

All animal experiments were performed in compliance with the German animal protection law (TierSchG BGBI S. 1105;

25.05.1998). The mice were handled in accordance with good animal practice as defined by FELASA and GV-SOLAS. All animal experiments were approved by the responsible state office (Lower Saxony State Office of Consumer Protection and Food Safety) under permit numbers #33.9-42502-04-12/0930 and #33.19-42502-04-17/2657. For NK cell depletion experiments, all animals were housed and bred under specific pathogen–free conditions at the Central Animal Facility, Faculty of Medicine, University of Rijeka, Croatia, in accordance with the guidelines contained in the International Guiding Principles for Biomedical Research Involving Animals. The Ethics Committee at the University of Rijeka and National ethics committee approved all animal experiments.

## Mice

Mice used at the animal facility of the Helmholtz Centre for Infection Research in Braunschweig, Germany, and at the Central Animal Facility, Faculty of Medicine, University of Rijeka, Croatia, were bred and maintained under specific pathogen-free conditions. STING knockout (MPYS$^{-/-}$/Tmem173$^{tm1.2Camb}$) and *goldenticket* (C57BL/6J-Tmem173$^{gt}$) mice have been described (Jin *et al*, 2011; Sauer *et al*, 2011).

C57BL/6J (B6J) or STING$^{-/-}$ mice were infected by the intravenous route with $4 \times 10^5$ PFU. Balb/c mice were infected by the intravenous route with $1 \times 10^6$ PFU. Six hpi serum and spleen were collected and analyzed for cytokines and viral transcripts.

NK cells were depleted from B6J or STING$^{-/-}$ mice with α-NK1.1 (clone PK136, 250 µg per mouse, intraperitoneal route) 1 day prior infection. Efficacy of NK cell depletion was analyzed by cytofluorometric analysis 1 day after depletion. Mice were then infected with MCMV as described above and spleen and serum were collected 16 hpi.

## Viruses

Manipulation of the MCMV genome was carried out by *en passant* mutagenesis (Tischer *et al*, 2010) on the MCK-2 repaired MCMV BAC-plasmid (Jordan *et al*, 2011) in which the m157 ORF has been deleted (Δm157). pEP-KanS (Tischer *et al*, 2010) served as the template for PCR. For construction of the recombinant MCMV-m152stop, a linear PCR product was generated using primers m152STOPEPfor: 5′-<u>CTACTTGCTCCTCTCGGTTC TCATAAACCGA GGCGAGACG</u>**GGCTAGTTAACTAGCC**<u>GCGGGCAGCAGCTATATGGA AG</u>GATGACGACGATAAGTAGGG-3′ and m152STOPEPrev: 5′-<u>CATCCTCGAATATGCGCACGTCCATATAGCTGCTGCCCGC</u>**GGCTAG TTAACTAGCC**<u>CGTCTCGCCTCGGTTTATGA</u>*CAACCAATTAACCAAT TCTGATTA*-3′ to introduce a stop cassette (bold) between nucleotide positions 211,421 and 211,422 (accession #GU305914). MCMV-specific sequences are underlined. The recombinant MCMV BAC m152stop was fully sequenced by pair-end sequencing using the MiSeq System (Illumina) to verify that no errors were introduced compared to MCMV BAC Δm157. Recombinant MCMV BACs were reconstituted and high titer virus stocks prepared as described previously (Chan *et al*, 2017). Unless otherwise stated, MCMV Δm157 is designated as parental MCMV. Balb/c mice were infected with MW97.01 (Wagner *et al*, 1999) and the corresponding Δm152 MCMV (Wagner *et al*, 2002).

Newcastle disease virus (NDV) was kindly provided by Andrea Kröger (Helmholtz Centre for Infection Research, Braunschweig, Germany).

## Cell lines

M2-10B4 (CRL-1972), HeLa S3 (CCL-2.2), and human embryonic kidney 293-T/17 cells (293T, CRL-11268) were obtained from ATCC. 293T and M2-10B4 cells were maintained in Dulbecco's modified Eagle's medium (DMEM; high glucose) supplemented with 8% fetal calf serum (FCS), 2 mM glutamine (Gln), and 1% penicillin/strepto-mycin (P/S). HeLa S3 were cultured using the same medium described above, supplemented with 1% non-essential amino acids (NEAA) and 1 mM sodium pyruvate.

Primary B6J wild-type (WT) mouse embryonic fibroblasts (MEF) and *goldenticket* MEF (MEF$^{gt/gt}$) were immortalized by SV40 LT antigen. The immortalized murine bone marrow-derived macrophage (iBMDM) cell line was obtained through BEI Resources, NIAID NIH (NR-9456). iBMDM were maintained in DMEM supplemented with 8% FCS, 2 mM Gln, 1% P/S, and 50 μM β-mercaptoethanol. iMEF were maintained in the same medium as iBMDM with additional 1% NEAA. iBMDM and iMEF stably expressing empty vector (ev) or m152-V5/His were generated by retroviral transduction using the constructs pMSCVpuro and pMSCVpuro m152-V5/His and selection with 10 μg/ml puromycin. iMEF$^{gt/gt}$ stably expressing cherry-tagged WT STING or K288R STING were generated by retroviral transduction and sorted by FACS to select for Cherry-positive cells. For generation of iMEF$^{gt/gt}$ expressing Cherry-STING and either ev or m152-V5/His, iMEF$^{gt/gt}$ Cherry-STING were transduced with pMSCVpuro or pMSCVpuro m152-V5/His, respectively, and selection was carried out as described above.

## Primary cells

For generation of primary bone marrow-derived macrophages, bone marrow was extracted from wild-type B6J mice and cells were cultured in DMEM (high glucose) supplemented with 10% FCS, 2 mM Gln, 1% P/S, 50 μM β-mercaptoethanol, and 5% macrophage colony-stimulating factor (MCSF) as described previously (Bussey *et al*, 2014). Primary mouse embryonic fibroblasts from B6J and Balb/c mice were generated by standard protocol.

## Plasmids

Expression constructs for m152-V5/His, M35-V5/His, LacZ-V5/His, and M27-V5/His (all in pcDNA3.1-V5/His, Invitrogen) have been described previously (Munks *et al*, 2006). pBS-mCD4 was purchased from Addgene (#14613) and subcloned into pcDNA3.1-V5/His via the *HindIII/BstXI* sites to generate pcDNA3.1 mCD4-V5/His. pcDNA4 ORF36-myc/His was previously described (Bussey *et al*, 2014) and codes for ORF36 of Kaposi's sarcoma-associated herpesvirus (KSHV). pPolI A/California/04/2009 NS was kindly provided by Toru Takimoto (University of Rochester Medical Center, USA) and subcloned into pcDNA3.1(-) via the *NotI/EcoRI* sites to generate pcDNA3.1(-) Cal NS1.

pEFBOS mCherry-mSTING (designated Cherry-STING) expressing monomeric Cherry fused to the N terminus of murine STING and pIRESneo3 cGAS-GFP (GFP fused to the C terminus of human cGAS) were kindly provided by Andrea Ablasser (Global Health Institute, Ecole Polytechnique Fédérale de Lausanne, Switzerland). pcDNA3 hSTING coding for untagged human STING was described previously (Christensen *et al*, 2016). pRL-TK, which expresses

Renilla luciferase under control of the thymidine kinase promoter, and pIRES2-GFP were purchased from Promega and Clontech, respectively. pNF-κB Luc, containing five NF-κB responsive elements (TGGGGACTTTCCGC) upstream of the firefly luciferase gene, is commercially available from Agilent Technologies. pGL3basic IFNβ-Luc (IFNβ-Luc) and pGL3basic ISG56-Luc (ISG56-Luc) were described previously (Chan *et al*, 2017). The firefly luciferase reporter plasmids p125, consisting of the human IFNβ promoter region (−125 to +19), its related mutant p125AA Luc (CC to AA, −58) with disrupted NF-κB binding, and p55-CIB, containing 8 tandem repeat motifs (AAGTGA) corresponding to 7 repeats of an IRF binding element (AANNGAAA), were kindly provided by Takashi Fujita (Kyoto University, Japan) (Fujita *et al*, 1989; Yoneyama *et al*, 1996). pcDNA3-FLAG-TBK1 was described previously (Christensen *et al*, 2016). CMVBL IRF3-5D codes for human IRF3 containing five amino acid substitutions (S396D, S398D, S402D, S404D, S405D) which render it constitutively active and was provided by John Hiscott (Institut Pasteur Cenci Bolognetti Foundation, Rome, Italy). pCAGGS Flag-RIG-I N, expressing a constitutively active truncation mutant of RIG-I, was kindly provided by Andreas Pichlmair (Technical University Munich, Germany).

Cherry-STING and m152-V5/His were subcloned into pMSCVpuro (Clontech) via the *BglII/EcoRI* sites to generate pMSCVpuro Cherry-STING and pMSCVpuro m152-V5/His, respectively. pcDNA3.1-m152-N83Q-N230Q-N263Q (Fink *et al*, 2013), pSP64-poly(A)-β-actin (Wilhelmi *et al*, 2008), pSP64-poly(A)-IE1 (Kurz *et al*, 1999), and pDrive-E1 (Simon *et al*, 2006) were published previously. pSP64-poly(A)-m152 was generated by subcloning of the PCR amplified ORF m152 into pSP64-poly(A) via the *HindIII/XmaI* sites. pcDNA3.1-m152-N83Q-N230Q-N263Q was subcloned into pcDNA3.1-V5/His via the *BamHI/BstXI* sites to generate pcDNA3.1m152- N83Q-N230Q-N263Q-V5/His.

The following constructs were cloned using the Q5 site-directed mutagenesis kit (NEB #E0554) according to the manufacturer's protocol: pcDNA3.1m152-ΔCTD-V5/His was generated by deletion of the last 23 amino acids (aa) of the C terminus of m152 (accession #GU305914, region 210342-211478). For construction of pcDNA3.1m152-CD4CTD-V5/His, the last 25 aa of m152 were replaced with the 33 C-terminal aa of murine CD4 (mCD4, accession #NM_013488.2). pcDNA3.1m152-CD4TM-V5/His was generated by replacing aa 327–352 of m152 with aa 395–419 of mCD4. pcDNA3.1m152-CD4TMCTD-V5/His was constructed by exchanging aa 327–378 of m152 with aa 395–457 of mCD4. For pcDNA3.1 CD4-m152SPCTD-V5/His, aa 1–26 of mCD4 were replaced with aa 1–19 of m152 and aa 420–457 were replaced by the corresponding sequence of m152. For pcDNA3.1 CD4-m152SPTMCTD-V5/His, aa 1–26 of mCD4 were replaced with aa 1–19 of m152 and aa 395–457 were replaced by the corresponding sequence of m152. pcDNA3.1m152-Δstalk was generated by deletion of aa 300–326. pcDNA3.1m152-GSstalk was constructed by replacing aa 300–326 with a glycine-serine linker (G$_4$S)$_5$. For murine Cherry-STING, the N to E mutation at position 41 or the exchange of QNTADIY → PNAVGPP at position 109–115 was introduced singly or in combination. Human STING mutants were generated by introducing the E to N mutation at position 41 or the PNAVGPP → QNTADIY aa exchange at position 110–116 singly or in combination. Constitutively active murine STING was generated by introducing the mutation V154M. Murine Cherry-STING lysine mutants were generated by singly introducing the mutations K83R, K235R, K275R, K288R, or K337R. All constructs were verified by

sequencing. Primer sequences as well as sequences of all constructs are available upon request.

## Antibodies and reagents

Rabbit anti-STING (#13647, clone D2P2F), rabbit anti-phospho-TBK1 (#5483, clone D52C2) and rabbit anti-phospho-IRF3 (#4947, clone 4D4G), anti-p65 (#4764, clone C22B4) antibodies were purchased from Cell Signaling Technology. Mouse anti-tubulin (T6199, clone DM1A), mouse anti-actin (A5441, clone AC-15), and Brefeldin A (B7651) were obtained from Sigma-Aldrich. The mouse anti-V5 antibody was purchased from Invitrogen (R960-25) and Biolegend (#680601); the mouse anti-V5 HRP conjugate was from Invitrogen (R961-25). Mouse antibodies against MCMV IE1 (#HR-MCMV-08, clone CROMA101) and MCMV m152 (#HR-MCMV-11, clone m152.05) were generated at the Center for Proteomics (CapRi), Faculty of Medicine, University of Rijeka. HRP-conjugated and Alexa Fluor®-conjugated secondary antibodies were purchased from Dianova and Invitrogen, respectively.

2′3′-cGAMP (#tlrl-nacga23) and high molecular weight poly(I:C) (#tlrl-pic) were purchased from Invivogen. CpG-B DNA 1826 was obtained from Eurofins Genomics. Interferon-stimulatory DNA (ISD) was generated by the combination of complementary forward (ISD45 bp-for: 5′-TACAGATCTACTAGTGATCTATGACTGATCTGTA CATGATCTACA) and reverse (ISD45 bp-rev: 5′-TGTAGATCATGTA CAGATCAGTCATAGATCACTAGTAGATCTGTA) 45 bp oligonucleotides, heating to 70°C for 10 min followed by annealing at room temperature. Protease inhibitors (4693116001) and phosphatase inhibitors (4906837001) were purchased from Roche.

The transfection reagents Lipofectamine 2000, FuGENE HD, and polyethyleneimine (PEI) were purchased from Life Technologies, Promega, and Polysciences, Inc., respectively. OptiMEM was obtained from Thermo Fisher Scientific.

## Luciferase-based reporter assays

cGAS-STING: 293T cells (25,000 cells/well, 96-well format) were transiently transfected with 60 ng Cherry-STING, pcDNA3 hSTING, or STING mutants, 60 ng pIRESneo3 human cGAS-GFP (for stimulated conditions), 60 ng pIRES2-GFP (for unstimulated conditions), 100 ng pGL3basic IFNβ-Luc, 10 ng pRL-TK, 120 ng plasmid of interest, and 1.2 μl FuGENE HD (Promega) diluted to 10 μl total volume with OptiMEM. To analyze activation of other luciferase reporters, pGL3basic-IFNβ-Luc was substituted with 50 ng p125, 50 ng p125AA, or 1 ng of p55-CIB reporter plasmids. For NF-κB Luciferase assays, cells were transfected with 10 ng pNF-κB Luciferase and 20 ng pRL-TK instead of 100 ng pGL3basic IFNβ-Luc and 10 ng pRL-TK.

RIG-I N: 293T cells (25,000 cells/well, 96-well format) were transiently transfected with 13 ng pCAGGS Flag-RIG-I N (stimulated) or pcDNA3.1 (unstimulated) together with 50 ng pGL3basic IFNβ-Luc, 5 ng pRL-TK, and 130 ng plasmid of interest and 0.66 μl FuGENE HD (Promega) diluted to 10 μl total volume with OptiMEM.

TBK1 and IRF3-5D: 293T cells (25,000 cells/well, 96-well format) were transiently transfected with 10 μl of FuGENE HD/DNA complexes composed of 100 ng pcDNA3-FLAG-TBK1 or CMVBL IRF3-5D (stimulated) or 100 ng pIRES2-GFP (unstimulated), 100 ng pGL3basic IFNβ-Luc, 10 ng pRL-TK, 120 ng plasmid of interest, and 1 μl FuGENE HD (Promega) diluted in OptiMEM.

For the cGAS-STING, RIG-I N and IRF3-5D luciferase assays cells were lysed 20 h post-transfection in 1× passive lysis buffer (Promega).

IFNβ stimulation and ISG56 reporter assay: 293T cells (25.000 cells/well, 96-well format) were transiently transfected with 10 μl of FuGENE HD/DNA mixtures containing 100 ng pGL3basic ISG56-Luc, 10 ng pRL-TK, 120 ng plasmid of interest, and 0.8 μl FuGENE HD diluted in OptiMEM. 24 h post-transfection, cells were stimulated by addition of recombinant human IFNβ (PeproTech, #300-02BC) to a concentration of 0.1 ng/ml or were mock stimulated and lysed 16 h later in 1× passive lysis buffer (Promega).

Luciferase production was measured using the dual-luciferase assay system (Promega) and either a GloMax 96 Microplate Luminometer (Promega) or an Infinite M200 PRO plate reader (Tecan). Luciferase fold induction was calculated by dividing Renilla-normalized values from stimulated samples by the corresponding values from unstimulated samples.

## Quantitative RT-PCR

iMEF$^{gt/gt}$ stably expressing Cherry-STING and either ev or m152-V5 were stimulated by transfection with either 5 μg/ml ISD or 10 μg/ml poly(I:C) complexed with Lipofectamine, or transfected with Lipofectamine only. 4 hours post-transfection, cells were lysed in RLT buffer supplemented with β-mercaptoethanol and RNA was purified with the RNeasy Mini Kit (Qiagen #7410) followed by DNase treatment (Qiagen #79254) according to the manufacturer's instructions. For synthesis of cDNA and quantification of gene transcripts, 100 ng of RNA was used per sample and quantitative RT-PCR was performed using the EXPRESS One-Step Superscript™ qRT-PCR Kit (Invitrogen #11781200) on a LightCycler 96 instrument (Roche). *Rlp8* served as the housekeeping control. PCR primers and Universal probe library (UPL, Roche) probes used were as follows: Rlp8 (Rlp8_for: 5′ caaca gagccgttgttggt, Rlp8_rev: 5′ cagcctttaagataggcttgtca, UPL probe 5); IFNβ (IFNβ_for: 5′ ctggcttccatcatgaacaa, IFNβ_rev: 5′ agagggctgtggtg gagaa, UPL probe 18); IL-6 (IL6_for: gctaccaaactggatataatcagga, IL6_rev: ccaggtagctatggtactccagaa, UPL probe 6).

For the quantification of viral transcripts in the spleen, quantitative RT-PCR specific for MCMV m123/IE1, MCMV M112/E1, and MCMV m152 was performed. RNA was extracted from spleen homogenates using the RNeasy Mini Kit followed by DNase treatment as described above. Quantitative RT-PCR was performed using the OneStep RT-PCR Kit (Qiagen #210212) on a LightCycler 96 instrument (Roche). Absolute quantification of viral transcript numbers was performed using a dilution series of specific *in vitro* transcripts as standards. For normalization, cellular β-actin transcripts were quantified in parallel. For generation of the dilution series, pSP64 poly(A) β-actin, pSP64 poly(A) IE1, pDrive E1, and pSP64 poly(A) m152 were first linearized using *Eco*RI and *in vitro* transcribed using the MEGAscript T7 Kit (Thermo Fisher Scientific, #AM1333). The number of generated transcripts was calculated, and a serial dilution of transcripts with defined, graded amounts per microliter was prepared. PCR primers and probe (5′ 6-FAM labeled, 3′ black hole quencher (BHQ) labeled) sequences were as follows: β-actin: β-actin_for: 5′ gacggccaggtcatcac tattg, β-actin_rev: 5′ cacaggattccatacccaagaagg, β-actin_probe: 5′ aacgagcggttccgatgccc; MCMV IE1: IE1_for: 5′ tggctgattgatagttctgttt tatca, IE1_rev: 5′ ctcatggaccgcatcgct, IE1_probe: 5′ aacgctcctcact gcagcatgcttg; MCMV E1: E1_for: 5′ tgctcccactgaggaagagaaga, E1_rev:

5′ gaggccgctgctgtaacaat, E1_probe: 5′ agcccaagcgccagaagaccca; MCMV m152: m152_for: 5′ cgttcgcgagactgatgttgt, m152_rev: 5′ gcaacggctacgt gtcctgta, m152_probe: 5′ ccaacggaacctgagtgcgca.

## ELISA

IFNα levels were measured with a rat anti-mouse IFNα capture antibody (PBL #22100-1) and a rabbit anti-mouse IFNα detection antibody (PBL #32100-1). IFNβ production was detected using the PBL mouse IFNβ ELISA kit (PBL #42400-1) or the LumiKine mouse IFNβ ELISA kit (Invivogen #lumi-mifnb) according to the manufacturer's instructions. The IL-6 ELISA was performed according to the manufacturer's protocol (BD Biosciences #555240). TNFα levels were determined as described previously (Bussey *et al*, 2014). The preparation of spleens from infected mice for measuring type I IFN levels by ELISA has been described previously (Chan *et al*, 2017).

## Immunofluorescence

HeLa cells were seeded onto acid-washed coverslips in 24-well plates. 24 hours later, cells were transiently transfected with FuGENE HD/DNA complexes composed of 200 ng pEFBOS mCherry-mSTING, 200 ng pcDNA3.1m152-V5/His, either 150 ng pIRESneo3 human cGAS-GFP (stimulated) or 150 ng pcDNA3.1 (ev) (unstimulated) and 1.4 μl FuGENE HD (Promega) diluted in OptiMEM. iMEF$^{gt/gt}$ were similarly seeded and 24 h later transiently transfected with FuGENE HD/DNA complexes composed of 200 ng pcDNA3.1m152-V5/His together with either 200 ng pEFBOS mCherry-mSTING or 200 ng empty vector in combination with 200 ng pIRESneo3 human cGAS-GFP (stimulated) or 200 ng empty vector (unstimulated) and 2 μl FuGENE HD (Promega) diluted in OptiMEM. 293T cells were seeded in 24-well plates and transiently transfected with FuGENE HD/DNA complexes composed of 150 ng pEFBOS mCherry-STING, 300 ng V5-tagged CD4, m152, or the respective m152 mutant, together with either 150 ng pIRESneo3 human cGAS-GFP (stimulated) or 200 ng empty vector (unstimulated) and 2 μl FuGENE HD (Promega). To detect translocation of Cherry-STING WT or Cherry-STING K288R, 293T cells were transfected with FuGENE HD/DNA complexes composed of 200 ng pEFBOS mCherry-mSTING WT or pEFBOS mCherry-mSTING K288R in combination with 200 ng pIRESneo3 human cGAS-GFP (stimulated) or 200 ng empty vector (unstimulated) and 2 μl FuGENE HD (Promega) diluted in OptiMEM. iMEF$^{gt/gt}$, iMEF$^{gt/gt}$ stably expressing Cherry-STING WT or iMEF$^{gt/gt}$ stably expressing Cherry-STING K288R were seeded in 24-well plates and either stimulated by transfection with 10 μg/ml poly(I:C) complexed with Lipofectamine or infected with parental MCMV at an MOI of 0.05.

24 hours post-transfection or 4 h post-stimulation or infection, cells were permeabilized with ice-cold methanol for 5 min at −20°C followed by fixation with 4% PFA in PBS for 20 min at RT. Cells were washed three times with PBS and then incubated in 10% FCS and 1% BSA in PBS for 1 h at room temperature. Primary antibodies diluted in 1% BSA in PBS were added overnight at 4°C, followed by three PBS washes. Cells were incubated with secondary antibodies coupled to Alexa488, Alexa594, or Alexa647 and Hoechst (Thermo Fisher Scientific, #33342) diluted in 1% BSA in PBS for 45 min at room temperature. Coverslips were mounted on glass microscope

slides with Prolong Gold (Invitrogen). Imaging was performed on a Nikon ECLIPSE Ti-E-inverted microscope equipped with a spinning disk device (Perkin Elmer Ultraview), and images were processed using Volocity software (version 6.2.1, Perkin Elmer).

## Immunoblotting

For co-immunoprecipitation (co-IP) experiments in 293T cells, $6 \times 10^5$ cells were transiently transfected with 4 μg total DNA complexed with 15 μl PEI and diluted to 300 μl with OptiMEM. 24 hours post-stimulation, cells were lysed in 1% NP-40 lysis buffer (50 mM Tris–HCl pH 7.4, 150 mM NaCl, 0.5 mM EDTA, 1% IGEPAL CA-630 (NP-40 replacement)). Protease and phosphatase inhibitors were added freshly to all lysis buffers prior to use.

iMEF stably expressing ev or m152-V5 or iMEF$^{gt/gt}$ stably expressing Cherry-STING and either V5-tagged m152 or the corresponding control cells ($2.5 \times 10^5$ cells/well, 6-well format) were stimulated by transfection with 5 or 10 μg/ml ISD complexed with Lipofectamine and lysed in radioimmunoprecipitation (RIPA) buffer (20 mM Tris–HCl pH 7.5, 1 mM EDTA, 100 mM NaCl, 1% Triton X-100, 0.5% sodium deoxycholate, 0.1% SDS). For the detection of STING dimers, a modified lysis buffer was used consisting of 25 mM Tris–HCl pH 7.6, 150 mM NaCl, 1% IGELPAL CA-630, 1% sodium deoxycholate, and 0.1% SDS. When using Brefeldin A (BFA), an inhibitor for protein transport from the ER to the Golgi compartment, cells were pretreated with BFA (10 μg/ml) for 60 min prior to stimulation and during the stimulation. iMEF WT or iMEF$^{gt/gt}$ ($2.5 \times 10^5$ cells/well, 6-well format) were infected with parental MCMV or MCMV m152stop at an MOI of 0.5 and infection was enhanced by centrifugation at $805 \times g$ at 4°C for 30 min. For the analysis of STING signaling kinetics upon MCMV infection, iMEF WT ($2.5 \times 10^5$ cells/well, 6-well format) were infected with parental MCMV or MCMV m152stop at an MOI of 0.1 and the infection was enhanced by centrifugation at $805 \times g$ at 4°C for 30 min. After centrifugation (defined as time point 0), cells were incubated at 37°C and 7.5% $CO_2$ for 30 min followed by replacement of virus-containing medium with fresh medium. Cells were lysed at indicated time points in 1% NP-40 lysis buffer as stated above.

For co-IPs, one-tenth of the lysate was reserved as input lysate, and the remainder was pre-cleared with protein A agarose beads (IPA300, Repligen). Cleared lysates were then incubated with indicated antibodies overnight at 4°C, and protein A agarose beads were added for 1 h. Beads were washed at least 6 times with 1% NP-40 lysis buffer and bound protein was eluted by heating samples in SDS sample buffer. Input lysates and IP samples were then analyzed by immunoblotting.

Cell lysates and samples were separated by SDS–PAGE and transferred to nitrocellulose or PVDF membrane (both GE Healthcare) using wet transfer and Towbin blotting buffer (25 mM Tris, 192 mM glycine, 20% (v/v) methanol (pH 8.3)). Membranes were probed with the indicated primary antibodies and respective secondary HRP-coupled antibodies diluted in 5% w/v non-fat dry milk or 5% BSA in TBS-T. Immunoblots were developed with Lumi-Light (Roche) or SuperSignal West Femto (Thermo Fisher Scientific) chemiluminescence substrates. Membranes were exposed to films or imaged with a ChemoStar ECL Imager (INTAS). Images were prepared using Adobe Photoshop CS5.

## Live cell imaging

Live cell imaging was performed on a Nikon ECLIPSE Ti-E-inverted microscope equipped with a spinning disk device (Perkin Elmer Ultraview), and movies were recorded and processed using Volocity software (Perkin Elmer).

iMEF$^{gt/gt}$ were seeded onto a 8-well chamber slide (Sarstedt # 94.6190.802) for imaging. On the next day, iMEF$^{gt/gt}$ stably expressing Cherry-STING and ev were infected with parental MCMV or MCMV m152stop at an MOI of 0.5 and infection was enhanced by centrifugation at 805 × $g$ at 4°C for 30 min. For each virus, two wells were infected. After centrifugation (defined as time point 0), the chamber slide was placed onto the microscope housed in a humidified $CO_2$ chamber pre-heated to 37°C. For each experiment, 5 views were randomly selected per well for imaging. 45 min post-infection, virus-containing medium was replaced with fresh medium prior to the commencement of live cell imaging/capture. iMEF$^{gt/gt}$ stably expressing Cherry-STING and either ev or V5-tagged m152 were stimulated by transfection of 5 μg/ml ISD complexed with Lipofectamine. For each cell line, two wells were stimulated. Imaging was set up as described above and initiated after addition of ISD to the cells.

Cells were imaged for up to 4 h and STING translocation was quantified 120 min and 180 min post-ISD stimulation or infection. To quantify STING translocation, cells with a distinct STING perinuclear localization were counted and expressed as the percentage of the total number of cells per view. Each data point represents one live cell imaging movie of a single viewpoint, with a minimum of 75 cells captured per view.

## Statistical analysis

Differences between two data sets were evaluated by Student's *t*-test (unpaired, two-tailed), in the case of viral transcript levels after log transformation of the data sets, using GraphPad Prism version 5.0 (GraphPad Software, San Diego, CA). *P*-values < 0.05 were considered statistically significant.

**Expanded View** for this article is available online.

## Acknowledgements
We thank Vladimir Gonçalves Magalhães and Kendra A. Bussey for fruitful discussions and Georg Wolf and Christine Standfuβ-Gabisch for excellent technical assistance. This work has been carried out within the framework of the SMART BIOTECS alliance between the Technische Universität Braunschweig and the Leibniz Universität Hannover. This initiative is supported by the Ministry of Science and Culture (MWK) of Lower Saxony, Germany. This study was also funded by the Deutsche Forschungsgemeinschaft (DFG), BR3432/3-1, and the Helmholtz Gemeinschaft, Virtual Institute VISTRIE, VH-VI-424. NAWL was funded by the Deutsche Forschungsgemeinschaft, SFB 1292, individual project TP11–viral evasion of innate and adaptive immune cells and "inbetweeners." VJL and AK were supported by the grant "Strengthening the capacity of the Scientific Centre of Excellence CerVirVac for research in viral immunology and vaccinology", KK.01.1.1.01.0006, financed by the European Regional Development Fund".

## Author contributions
Conceptualization, MS, BC, MMB; Methodology, MS, BC, NAWL; Investigation, MS, BC, VJL, AK, JH, NF; Writing—Original Draft, MS, BC, MMB; Writing— Review & Editing, MS, BC, VJL, AK; JH, NF, SRP, NAWL, MMB; Funding Acquisition, MMB; Resources, VJL, AK, SRP, NAWL, MMB; Supervision, MS, BC, MMB.

## Conflict of interest
The authors declare that they have no conflict of interest.

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
