## [Review Process File · The EMBO Journal]

The herpesviral antagonist m152 reveals differential activation of STING-dependent IRF and NF- κ B signaling and STING's dual role during MCMV infection

Markus Stempel, Baca Chan, Vanda Juranić Lisnić, Astrid Krmpotić, Josephine Hartung, Søren R. Paludan, Nadia Füllbrunn, Niels A. W. Lemmermann, Melanie M. Brinkmann

Review timeline:

Submission date:	22nd Oct 2018
Editorial Decision:	19th Nov 2018
Revision received:	2nd Dec 2018
Editorial Decision:	12th Dec 2018
Revision received:	16th Dec 2018
Accepted:	19th Dec 2018

Editor: Karin Dumstrei

Transaction Report:

1st Editorial Decision

19th Nov 2018

Thank you for submitting your manuscript to The EMBO Journal. Your study has now been reviewed by two referees and their comments are provided below.

As you can see the referees appreciate that the analysis adds important new insight and support publication here. They raise some issues that I anticipate that you should be able to sort out in a good way. Given the concerns raised I would like to invite you to submit a suitably revised manuscript.

REFeree REPORTS:

Referee #1:

In the presented work, Brinkmann and colleagues reveal m152, a murine CMV expressed protein, as a natural antagonist of STING. The screening of a collection of MCMV ORFs led to the identification of m152, which upon overexpression, counteracted the STING-induced type I IFN response, but not the one induced by other pattern recognition receptors (e.g., RIG-I, TLR9). Further analysis shows that m152 interacts with STING at the endoplasmic reticulum and that this interaction is mediated by the N-terminal part of m152 and luminal regions of mouse but not human STING. As an effect, the translocation of STING to the Golgi - a prerequisite for type I IFN induction - is delayed concomitant with a delayed activation of the downstream signaling molecules TBK1 and IRF3, respectively. Interestingly, studying the role of m152 during MCMV infection, the investigators observed a selective impairment of m152 on STING-mediated type I IFN induction, but not on IL6 expression. This led to the hypothesis that STING may differentially regulate type I/IRF3 induction versus IL6/NF- κ B activation. Assessing several STING mutants for their effect on

IRF3 versus NF- κ B activity, the authors indeed identify one mutant construct (K288R) that failed to translocate and to trigger IFN β expression, but still exerted NF- κ B activity. Finally, it is shown that the selective induction of NF- κ B by the STING K288R mutant led to increased viral replication, which may explain why STING exerting both antiviral type I IFNs and proviral NF- κ B cannot confer resistance to viral replication in the context of MCMV.

Overall, this is a very interesting study, which uncovers a new means of viral antagonism targeting the cGAS-STING pathway. The observation that m152 differentially suppresses type I IFNs from proinflammatory cytokines is in particular intriguing. There are only minor comments that I would have.

1) Using a STING mutant construct, the authors show that NF- κ B signaling downstream of the STING pathway can be initiated from the ER. Does the NF- κ B activation remain intact if blocking ER-to-Golgi trafficking of STING (e.g., using compounds such as Brefeldin A or Nocodazole)?

2) Along these lines, is there evidence that NF- κ B activation occurs earlier upon STING activation than IRF-3?

3) The species-specific effect of m152 is interesting. Could one speculate about a similar mechanisms acting on human STING, which may be encoded by a human pathogenic herpes virus?

Referee #2:

EMBOJ-2018-100983
Stempel et al.

The multifaceted viral antagonist m152 reveals a dual role for STING during MCMV infection

Summary

Authors reveal that the murine cytomegalovirus (MCMV) m152 gene product, which has previously been shown to be a potent inhibitor of NK and T-cell by down regulation of RAE-1 and MHC class 1, is a direct inhibitor of the innate immune signaling molecule STING. They show that m152 binds to and inhibits murine, but not human, STING via its ER luminal domains to delay the translocation of STING to the golgi compartment where it drives the production of type I interferon. It does so independently of STING activation or dimerization. Due to this unique mechanism of inhibition, STING-dependent NF κ B activation remains intact to augment initial viral transcription, while type I interferon signaling is prevented.

Impressions

The major claim of this manuscript is that m152 is a multifunctional protein that targets STING during the early phases of infection. The authors demonstrate a very novel mechanism of action for m152, and provide compelling data to support their claims. For the most part, the work is high quality.

Major criticisms

1) The mutant virus constructed, as well as the virus designated as 'parental' are also deficient for m157 (fig 6). While technically controlled throughout the manuscript, it's kind of disingenuous to be doing in vivo experiments in B6J mice and chalking the phenotype entirely up to m152. I can entirely appreciate that the cell culture experiments and transfections are supported by the data, but at best authors should be prefacing their conclusions with "in the absence of m157....." Moreover, it's also insufficient to suggest that loss of m157 influences MCMV replication (e.g. line 365-6 "Defective replication of MCMV lacking m152 in vivo is mediated by STING-dependent signaling") when the data provided is only transcriptional analyses. If you say replication, measure replication. You've already biased the in vivo infections in B6J mice by using an m157-null mutant, so there should be plenty of replication to measure.

2) What kinetic class is m152? Authors show expression at 2hrs post-infection. Is this an immediate-early gene? Pinning that down would REALLY help the model.

Minor Criticisms

- 1) Figure 1 B, C, D, E, J, K are all negative data, and could be more appropriate as supplementary data. Also, Fig 1 F&G could be combined.
- 2) Figure 3D - also could be supplementary.
- 3) Figure 4F - there's still a pretty decent amount of human STING co-IPing with m152, enough that it makes it tough to make really definitive statements about the amount being pulled down. Can the three independent experiments be quantitated and graphed to bolster the conclusion?
- 4) Figure 6 -Infection of m152-V5 transduced cells would be expected to rescue the defects seen with the knockout virus? This might help get at Major point #1.
- 5) Line 437 - looks like a space missing in "IRFarms." Maybe just my copy.

1st Revision - authors' response

2nd Dec 2018

Please see next page.

Response to reviewers

Referee #1:

In the presented work, Brinkmann and colleagues reveal m152, a murine CMV expressed protein, as a natural antagonist of STING. The screening of a collection of MCMV ORFs led to the identification of m152, which upon overexpression, counteracted the STING-induced type I IFN response, but not the one induced by other pattern recognition receptors (e.g., RIG-I, TLR9). Further analysis shows that m152 interacts with STING at the endoplasmic reticulum and that this interaction is mediated by the N-terminal part of m152 and luminal regions of mouse but not human STING. As an effect, the translocation of STING to the Golgi - a prerequisite for type I IFN induction - is delayed concomitant with a delayed activation of the downstream signaling molecules TBK1 and IRF3, respectively. Interestingly, studying the role of m152 during MCMV infection, the investigators observed a selective impairment of m152 on STING-mediated type I IFN induction, but not on IL6 expression. This led to the hypothesis that STING may differentially regulate type I/IRF3 induction versus IL6/NF- κ B activation. Assessing several STING mutants for their effect on IRF3 versus NF- κ B activity, the authors indeed identify one mutant construct (K288R) that failed to translocate and to trigger IFN β expression, but still exerted NF- κ B activity. Finally, it is shown that the selective induction of NF- κ B by the STING K288R mutant led to increased viral replication, which may explain why STING exerting both antiviral type I IFNs and proviral NF- κ B cannot confer resistance to viral replication in the context of MCMV.

Overall, this is a very interesting study, which uncovers a new means of viral antagonism targeting the cGAS-STING pathway. The observation that m152 differentially suppresses type I IFNs from proinflammatory cytokines is in particular intriguing. There are only minor comments that I would have.

We thank the reviewer for these very nice words of appreciation.

1) Using a STING mutant construct, the authors show that NF- κ B signaling downstream of the STING pathway can be initiated from the ER. Does the NF- κ B activation remain intact if blocking ER-to-Golgi trafficking of STING (e.g., using compounds such as Brefeldin A or Nocodazole)?

Thank you for this insightful comment. Indeed, we have performed this experiment and used Brefeldin A to block ER to Golgi transport. However, in the presence of Brefeldin A, we did not observe STING-dependent activation of the NF- κ B pathway (and, as expected and shown in **Figure EV3B**, of the IRF pathway). This result was not completely surprising for us, since Brefeldin A leads to a collapse of the Golgi compartment, and results in accumulation of proteins in the ER, which may negatively affect their function. This result therefore neither supported nor contradicted our hypothesis that STING may activate NF- κ B signaling already from the ER. Since the Brefeldin A approach did not give us a conclusive result, we performed the screen for a STING mutant that is unable to leave the ER and can therefore not induce the interferon response. With the K288R mutation we could identify such a mutant and test if it was capable of inducing the NF- κ B pathway, which it could. This approach allowed us to draw the conclusion that STING indeed activates the NF- κ B pathway from the ER.

2) Along these lines, is there evidence that NF- κ B activation occurs earlier upon STING activation than IRF-3?

To our knowledge, no data exists in the literature about the precise kinetics of NF- κ B and IRF3 activation upon STING signaling. We have not performed a time course experiment (using immunofluorescence or cellular fractionation and immunoblotting as a readout) to determine NF- κ B and IRF3 nuclear translocation upon activation of the cGAS-STING pathway. The reasons are the following: We do not know how much of the total NF- κ B pool in the cell is activated by ER-residing STING, however, we see that whatever gets activated suffices to fully activate NF- κ B mediated transcription (**Figure 8D**). Since NF- κ B is a well expressed protein, nuclear translocation of small amounts of NF- κ B are difficult to be visualized - a faint signal in the nucleus may be masked by the strong signal in the cytosol, and we do not know how much NF- κ B actually needs to be present in the nucleus to see transcriptional activation. In addition, antibodies have very different specificities and sensitivities, and just because we detect an endogenous protein by immunoblotting or immunofluorescence at a specific timepoint, it does not mean that it may not be expressed or present in a certain compartment already at an earlier timepoint. This is why we think that it would be difficult to draw conclusions from a time course experiment using immunofluorescence or cellular fractionation and compare NF- κ B with IRF3. Even looking at phosphorylated versions of p65 or IRF3 by immunoblotting in a time-resolved manner may not be conclusive due to different sensitivities of the respective antibodies.

When looking into the literature regarding STING-dependent signaling, not much has been shown regarding STING-dependent NF- κ B activation. However, there is evidence that STING activates NF- κ B via a different mechanism than the activation of IRF3 by the kinase TBK1, including the publication from the Barber group showing that the human STING K224R mutant, which is not able to activate IRF signaling, can activate NF- κ B signaling (Ni et al., 2017). Our study with the murine STING K288R mutant shows that NF- κ B can be activated by ER-resident STING upon stimulation prior to STING trafficking (which is a prerequisite for IRF3 activation). Therefore we conclude that NF- κ B activation occurs at an earlier time point than IRF3 activation.

Along these lines, in the revised version of the manuscript we now include data showing that STING WT and STING K288R both still activated the NF- κ B reporter in TBK1 knockdown cells, but not the IRF3 reporter. We have now included this result as our new **Appendix Figure 3**. Taken together, we are convinced that STING-dependent NF- κ B activation occurs prior to trafficking of STING to the Golgi compartment, and consequently prior to activation of IRF3, and that this NF- κ B activation is independent of the kinase TBK1. However, how STING exactly activates the NF- κ B pathway from the ER still merits further investigation.

3) The species-specific effect of m152 is interesting. Could one speculate about a similar mechanisms acting on human STING, which may be encoded by a human pathogenic herpes virus?

This is indeed an interesting thought. To our knowledge, no such mechanism has been reported about for a human herpesvirus. Since herpesviruses have evolved with their respective hosts, a similar or comparable mechanism can, but not necessarily has to, be present in a human beta-herpesvirus. For example, human cytomegalovirus encodes the UL82 protein, which targets trafficking of STING, but by a different mechanism than m152 (Fu et al., 2017, Cell Host & Microbe). The interaction of UL82 and STING disrupts both IRF3- and NF- κ B dependent responses, while m152 interacts with STING via its luminal loop regions and does not disturb the cytosolic signaling platform, and thus NF- κ B signaling can still occur. While HCMV and MCMV show a high degree of similarity, they may have different requirements at different stages of their respective life cycle. Another possibility is that HCMV has evolved other means to activate the NF- κ B response to promote viral replication.

Referee #2:

EMBOJ-2018-100983

Stempel et al.

The multifaceted viral antagonist m152 reveals a dual role for STING during MCMV infection

Summary

Authors reveal that the murine cytomegalovirus (MCMV) m152 gene product, which has previously been shown to be a potent inhibitor of NK and T-cell by down regulation of RAE-1 and MHC class 1, is a direct inhibitor of the innate immune signaling molecule STING. They show that m152 binds to and inhibits murine, but not human, STING via its ER luminal domains to delay the translocation of STING to the golgi compartment where it drives the production of type I interferon. It does so independently of STING activation or dimerization. Due to this unique mechanism of inhibition, STING-dependent NFκB activation remains intact to augment initial viral transcription, while type I interferon signaling is prevented.

Impressions

The major claim of this manuscript is that m152 is a multifunctional protein that targets STING during the early phases of infection. The authors demonstrate a very novel mechanism of action for m152, and provide compelling data to support their claims. For the most part, the work is high quality.

Thank you very much for taking the time to carefully assess our manuscript and your very constructive criticism. We are very pleased that you consider our study novel and compelling.

Major criticisms

1) The mutant virus constructed, as well as the virus designated as 'parental' are also deficient for m157 (fig 6). While technically controlled throughout the manuscript, it's kind of disingenuous to be doing in vivo experiments in B6J mice and chalking the phenotype entirely up to m152. I can entirely appreciate that the cell culture experiments and transfections are supported by the data, but at best authors should be prefacing their conclusions with "in the absence of m157....." Moreover, it's also insufficient to suggest that loss of m157 influences MCMV replication (e.g. line 365-6 "Defective replication of MCMV lacking m152 in vivo is mediated by STING-dependent signaling") when the data provided is only transcriptional analyses. If you say replication, measure replication. You've already biased the in vivo infections in B6J mice by using an m157-null mutant, so there should be plenty of replication to measure.

Thank you for this valuable comment.

Regarding the term "replication": we agree and have modified the text accordingly. As pointed out by the reviewer, we have only measured transcription of viral genes and not replication by standard plaque assay. We could not measure replication because we look at very early time points (6 and 16h) post infection. At this point, the virus has not replicated enough to measure viral titers. It was not possible to infect the mice with higher viral doses, they would have succumbed to it. We cannot look at later time points because m152 also has an effect on the NK cell response and CD8 T cell response at later stages of infection. Moreover, the type I IFN response at this early phase of infection (6-16h) is STING-dependent - at later time points (e.g. 36h post infection) TLRs are responsible for the type I IFN response and not STING. Since m152 only affects STING-mediated type I IFN induction we therefore have to look at those early time points.

Regarding the MCMV m157 protein: we kindly but firmly disagree. We think that we can draw the conclusion that the phenotype we observe is solely due to the m152 protein, and not to the m157 protein. As shown in Figure S6 (**now Figure EV5**), we observe the same phenotype as reported by us for B6J mice also in Balb/c mice infected with WT MCMV, which expresses both the m152 and the m157 protein, and the Δ m152 mutant, which does not express m152 but still expresses m157. We agree with the reviewer that performing the *in vivo* experiments in B6J mice with the Δ m157 virus was not optimal, but since STING^{-/-} mice are not available on the Balb/c background, this was the only option. To convince ourselves that m157 does not contribute to the observed phenotype, we performed the Balb/c experiment shown in Figure S6 of our original submission (**now Figure EV5**). To make this more clear, we could consider moving current Figure EV5 into the main figures, but would leave this decision to the editor.

2) What kinetic class is m152? Authors show expression at 2hrs post-infection. Is this an immediate-early gene? Pinning that down would REALLY help the model.

Thank you for pointing this out. Indeed, we see very early expression of m152 in the context of MCMV infection by qPCR and immunoblotting (**Figure EV4**). Previous studies on m152 have proposed that m152 is expressed with early kinetics. However, to our knowledge, no study has investigated the m152 expression pattern according to the classical definition of the herpesviral expression stages.

We now performed the experiment that formally addresses the question if m152 expression occurs with immediate-early or early kinetics:

iMEF were infected with parental MCMV (MCMV Δ m157) at an MOI of 0.1 with centrifugal enhancement. Kinetic phases of viral gene expression, referred to as immediate-early (IE), early (E) and late (L) were determined as follows:

(i) immediate-early phase: 15 minutes prior to MCMV infection, culture medium was replaced with fresh medium containing 50 μ g/ml Cycloheximide (CHX, stock solution diluted in DMSO) to block *de novo* protein synthesis. Infection was performed in the presence of CHX. 3 hours post infection, cells were washed with medium containing 5 μ g/ml Actinomycin D and were incubated for 2 hours to allow IE gene expression and block further E-phase transcription.

(ii) early phase: The culture medium was replaced with fresh medium containing 250 μ g/ml phosphonoacetic acid (PAA) 15 minutes before infection to block DNA synthesis and thereby MCMV replication, allowing the expression of IE and E genes, but not L proteins. MCMV infection was performed in the presence of PAA and cells were harvested at 16 hours post infection.

(iii) late phase: The culture medium was replaced with fresh medium without the addition of metabolic inhibitors 15 minutes before infection. Cells were harvested at 24 hours post infection.

Samples were analyzed using immunoblotting analysis to determine the protein expression of immediate-early (MCMV IE1), early (MCMV M45) and late (MCMV M55) proteins.

Representative result of two independent experiments:

Based on our results, m152 would be classified as an early gene according to the classical definition of herpesviral expression kinetics. However, this classical definition is dependent on the aforementioned experimental setup, which may or may not reflect the real picture. In 2014, Weekes et al. published quantitative temporal viromics for HCMV and observed that the IE/E/L classification is not descriptive enough and proposed a model of five temporal classes for gene expression. This may also apply to MCMV.

Supporting our results, Marcinowski et al. (2012) reported transcriptomic profiling during MCMV infection and showed that newly synthesized m152 mRNA reaches its peak already at 1-2 hours post MCMV infection. We also see this in our experimental setups.

Taken together, we conclude that although MCMV m152 falls into the classical category of an early gene, the very early transcription of the m152 gene upon infection enables MCMV to express the protein at very early time points post MCMV infection.

Minor Criticisms

1) Figure 1 B, C, D, E, J, K are all negative data, and could be more appropriate as supplementary data. Also, Fig 1 F&G could be combined.

Thank you for this comment. However, we would prefer to leave these figures in the main figure. Although they are “negative” results, they are important to demonstrate the specificity of m152. But we leave the final decision to the editor.

2) Figure 3D - also could be supplementary.

We agree; we moved Figure 3D now to **EV Figure 2A**.

3) Figure 4F - there's still a pretty decent amount of human STING co-IPing with m152, enough that it makes it tough to make really definitive statements about the amount being pulled down. Can the three independent experiments be quantitated and graphed to bolster the conclusion?

We now quantified the three independent experiments and graphed the results as suggested. The results are presented as new **Appendix Figure 2B** and support our conclusions.

4) Figure 6 -Infection of m152-V5 transduced cells would be expected to rescue the defects seen with the knockout virus? This might help get at Major point #1.

Thank you for this valuable comment. Indeed, when infecting cells stably expressing m152-V5 with MCMV m152stop, we would expect a rescue of the phenotype. However, as pointed out above, since we performed and thoroughly controlled our *in vitro* and *in vivo* experiments, we can clearly pinpoint the phenotype down to the m152 protein.

5) Line 437 - looks like a space missing in "IRFarms." Maybe just my copy.

Thank you, we have corrected this.

Thank you for submitting your manuscript to The EMBO Journal. Your study has now been re-reviewed by the referees and their comments are provided below. As you can see both referees appreciate the introduced changes and support publication here. I am therefore very pleased to accept the manuscript for publication in The EMBO Journal.

Before I can send you the formal acceptance letter there are just a few things that we have to sort out.

- Please add 3-5 keywords to the manuscript
- Please add a running title to the manuscript
- We are also missing an AC (Author contributions) section.
- The figures should be removed from main MS file but their legends should stay in. The EV figure legends need to be added to main MS.
- You have at the moment 7 EV figures, but we can only have 5. Is there a way to combine some of them or alternatively you can add the 2 extra ones to the appendix.
- Figure callouts: Fig 3D is not called out; Fig 3F is called out but there is no Fig 3F.
- The M&M should be placed before References.
- The appendix file needs a Table of content
- Our publisher has done a pre-publication check on the manuscript and has some comments (see figure legends). Please address their comments and incorporate their suggestions. You should be able to see the file when you log in. The file is called Wiley Pre-acceptance Check - please check the word document as it is this file where you can see the marked changes
- We now encourage the publication of source data, particularly for electrophoretic gels and blots, with the aim of making primary data more accessible and transparent to the reader. It would be great if you could provide me with a PDF file per figure that contains the original, uncropped and unprocessed scans of all or key gels used in the figure? The PDF files should be labeled with the appropriate figure/panel number, and should have molecular weight markers; further annotation could be useful but is not essential. The PDF files will be published online with the article as supplementary "Source Data" files.
- We include a synopsis of the paper that is visible on the html file (see <http://emboj.embopress.org/>). Could you provide me with a general summary statement and 3-5 bullet points that capture the key findings of the paper?
- It would also be good if you could provide me with a summary figure that I can place in the synopsis. The size should be 550 wide by 400 high (pixels).

That should be it - you can use the link below to upload the revised version. Let me know if you have any further questions

REFEREE REPORTS:

Referee #1:

The authors` s reply to my concerns are convincing. I do not have any additional comments.

Referee #2:

The authors have sufficiently addressed the concerns of this reviewer. Incorporation of additional data strengthens the conclusions and enhances the manuscript. While we may 'kindly, but firmly disagree' on the execution of the viral genetics, the authors' position is on this point is widely accepted in the field. This reviewer is ultimately comfortable with the authors conclusions.

2nd Revision - authors' response

16th Dec 2018

The authors made all editorial changes.

Corresponding Author Name: Melanie M. Brinkmann

Journal Submitted to: The EMBO Journal

Manuscript Number: EMBOJ-2018-100983